# M³GPT: An Advanced Multimodal, Multitask Framework for Motion Comprehension and Generation

**Mingshuang Luo**[1,2,3]**, Ruibing Hou**[1]***, Zhuo Li**[4]**, Hong Chang**[1,3]**,**
**Zimo Liu**[2]**, Yaowei Wang**[2,5]**, Shiguang Shan**[1,3]

[1]Key Laboratory of Intelligent Information Processing of Chinese Academy of Sciences (CAS),
Institute of Computing Technology, CAS, China
[2]Peng Cheng Laboratory, China, [3]University of Chinese Academy of Sciences, China
[4]WeChat, Tencent Inc, [5]Harbin Institute of Technology, Shenzhen
`mingshuang.luo@vipl.ict.ac.cn,{houruibing,changhong,sgshan}@ict.ac.cn`
`albertzli@tencent.com,liuzm@pcl.ac.cn,wangyaowei@hit.edu.cn`

## Abstract

This paper presents M³GPT, an advanced **M**ultimodal, **M**ultitask framework for **M**otion comprehension and generation. M³GPT operates on three fundamental principles. The first focuses on creating a unified representation space for various motion-relevant modalities. We employ discrete vector quantization for multimodal conditional signals, such as text, music and motion/dance, enabling seamless integration into a large language model (LLM) with a single vocabulary. The second involves modeling motion generation directly in the raw motion space. This strategy circumvents the information loss associated with a discrete tokenizer, resulting in more detailed and comprehensive motion generation. Third, M³GPT learns to model the connections and synergies among various motion-relevant tasks. Text, the most familiar and well-understood modality for LLMs, is utilized as a bridge to establish connections between different motion tasks, facilitating mutual reinforcement. To our knowledge, M³GPT is the first model capable of comprehending and generating motions based on multiple signals. Extensive experiments highlight M³GPT's superior performance across various motion-relevant tasks and its powerful zero-shot generalization capabilities for extremely challenging tasks. Project page: `https://luomingshuang.github.io/M3GPT/`.

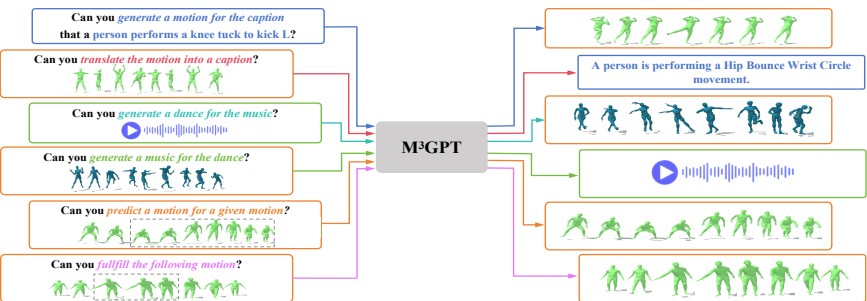

Figure 1: M³GPT can handle core motion comprehension and generation tasks, including text-to-motion, motion-to-text, music-to-dance, dance-to-music, motion prediction, and motion in-between. The motion sequences within the dashed-line areas are masked in the input.

---

*Corresponding author

38th Conference on Neural Information Processing Systems (NeurIPS 2024).

# 1   Introduction

Motion comprehension and generation in multimodality are crucial for diverse applications, including AR/VR creation, video games, and virtual reality. Numerous studies [15, 51, 64, 52] focus on motion comprehension, including captioning 3D human motions and generating music from 3D human dances[2]. Recent advancements in AI [14, 44, 42, 48] have paved the way for motion generation, allowing for various control signals including textual descriptions, music pieces, and human poses. A significant shortcoming of most existing works is their focus on single-modality control signals, overlooking the potential for multimodal information integration. More importantly, the comprehension and generation of motions are predominantly studied in isolation. In reality, human motion cognition and communication indispensably require seamless transitions between any motion-relevant modalities. Therefore, it is vital to develop a unified framework for motion comprehension and generation that can efficiently utilize multiple signals simultaneously.

Recent works [12, 62, 63, 58] have shown success in developing a unified multitask motion framework which integrates text-driven and audio-driven motion generation through a single architecture. Employing a large language model (LLM), [60] adeptly handles multimodal control signals, such as text and single-frame pose, to generate consecutive motions. Despite their promising performance in motion generation, these approaches often fall short in comprehending motion. MotionGPT [21], a recent innovation, constructs a unified motion-language model to generate plausible human motions and natural language descriptions through prompt instructions. However, MotionGPT focuses solely on a single non-motion modality, *i.e.*, text. While aligning motion with one additional modality is relatively straightforward, integrating three or more modalities within a single framework and achieving bidirectional alignment among them to cover a broad range of modalities for motion comprehension and generation presents a formidable challenge.

Two main challenges need to be solved for building a unified multimodal framework for motion comprehension and generation. ***The first is how to create a unified representation space across different motion-relevant modalities.*** MotionGPT [21] and SpeechGPT [54] separately treat motion and speech as specific language for seamlessly integrating with text. Inspired by these efforts [21, 54], we view both motion and music as distinct forms of language, facilitating better associations with text via LLMs. Specifically, akin to language, we compress raw motion and music into a sequence of discrete semantic tokens. By encoding motion, music, and language within a single vocabulary, we can build a unified representation space across these different modalities. ***The second is how to model the connections and synergies among various motion tasks.*** Different motion-relevant tasks are interconnected and can mutually enhance each other. Since text is the most familiar and well-understood modality for LLMs, we propose employing text as a bridge to establish connections between different motion tasks. Specifically, to better learn the complex music-to-dance task where both input and output modalities are unfamiliar to LLMs, we introduce two auxiliary tasks: music-to-text and text-to-dance, aimed at aligning music and dance modalities with the structured text embedding space. This strategy enables us to establish connections and synergies between music-to-dance and text-to-motion tasks, facilitating the alignment and collaboration of text, music, and motion/dance modalities across different tasks.

In this work, we propose a uniform **M**ultimodal, **M**ultitask framework for **M**otion comprehension and generation, namely M³GPT, that leverages the strong language generation capability of LLMs for unifying various motion-relevant tasks, as depicted in Fig. 1. M³GPT comprises three tires. **Firstly**, M³GPT is equipped with multimodal tokenizers capable of compressing raw multimodal data, including motion, music, and text, into a sequence of discrete semantic tokens. These discrete representations allow the core LLM to unify motion comprehension and generation in an autoregressive manner, operating at the discrete semantic representation space. **Secondly**, different from [21, 60] that solely optimize LLM in discrete semantic space, we jointly train LLM and motion de-tokenizer, optimizing LLM in both discrete semantic space and raw continuous motion space. This operation enables the motion-space error signals from de-tokenizer to backpropagate to LLM, enhancing LLM's ability to generate the details of motion. **Thirdly**, we construct paired text descriptions for music, and design two auxiliary music-to-text and text-to-dance tasks, which aid in aligning music and dance with the text embedding space. Also, we build up a shared tokenizer for motion and dance data

---

[2]In this paper, the term "motion" generally includes "dance." We distinguish them when referring to specific tasks or scenes, such as text-to-motion, and music-to-dance.

| Methods | T2M | M2T | A2D | D2A | M2M | Random M | Random T | Random A |
|---|---|---|---|---|---|---|---|---|
| TM2D[12] | ✔ | ✗ | ✔ | ✗ | ✗ | ✔ | ✗ | ✗ |
| UDE[62] | ✔ | ✗ | ✔ | ✗ | ✗ | ✔ | ✗ | ✗ |
| MotionGPT[60] | ✔ | ✗ | ✗ | ✗ | ✔ | ✔ | ✗ | ✗ |
| MotionGPT[21] | ✔ | ✔ | ✗ | ✗ | ✔ | ✔ | ✔ | ✗ |
| M$^3$GPT (Ours) | ✔ | ✔ | ✔ | ✔ | ✔ | ✔ | ✔ | ✔ |

Table 1: Comparison of recent multimodal, multitask methods across various motion comprehension and generation tasks. T2M: text-to-motion; M2T: motion-to-text; A2D: music-to-dance; D2A: dance-to-music; M2M: motion-to-motion that includes motion prediction and motion in-between. Random M, Random T, and Random A represent the unconstrained generation of motion, text, and music[3], respectively.

to project them into a shared semantic space. These auxiliary tasks and shared tokenizer establish connections between music-to-dance and text-to-motion, enabling mutual reinforcement.

We employ a multimodal pre-training + instruction-tuning pipeline to train M$^3$GPT, enhancing inter-modal alignment and effectively aligning them with human intent. To our knowledge, M$^3$GPT is the first approach to integrate six core tasks of motion comprehension and generation—text-to-motion, motion-to-text, music-to-dance, dance-to-music, motion prediction, and motion in-between—into a uniform framework. Extensive experiments demonstrate that M$^3$GPT achieves competitive performance across multiple motion-relevant tasks. Additionally, through qualitative results, we demonstrate that M$^3$GPT possesses powerful zero-shot generalization capabilities, *e.g*., long-term dance generation and music-text conditioned dance generation.

## 2 Related Work

**Motion comprehension and Generation.** Many existing works focus on studying human appearance, pose, detection, attribute, part parsing and so on [61, 19, 45, 40, 23, 17]. This work focuses on studying human motion, including motion comprehension and motion generation. Motion comprehension involves two core tasks: motion-to-text and dance-to-music. *Motion-to-text* aims to describe human motion with natural language [37]. For example, recurrent networks have been used in [37] to accomplish this task. *Dance-to-music* involves creating a piece of music from a given dance [20, 27, 64]. For example, Zhun *et al.* [64] utilizes a generative adversarial network to generate music from dance videos. On the other hand, motion generation involves generating diverse human motions using multimodal inputs, such as text [44, 56, 15, 5, 57], music [27, 18, 52, 42] and incomplete motion [31, 1, 3]. *Text-to-motion* is one of the most important motion generation tasks. Recent works typically map text to motion using different architectures: diffusion model [57] and autoregressive transformer model [15]. *Music-to-dance* focuses on generating dance movements from music. For example, [42] predicts discrete token sequences conditioned on music, which are then used to regenerate the dance sequence. *Motion Completion* generates motion conditioning on partial motions, such as motion prediction [31, 1] and motion-in-between [34, 43]. Although these methods have shown promising results in various human motion tasks, most are limited to handling a single task. Until recently, some works [12, 21, 60, 62] attempt to integrate two or more tasks into a unified model, as shown in Tab. 1. However, these works either lack the ability of motion comprehension [62, 60] or fail to handle music modality [12, 21]. In this work, we propose a unified motion comprehension and generation framework that can handle multiple control signals simultaneously.

**Language Models and Multimodal.** Large language models (LLMs) enabled by extensive datasets and model size, such as T5 [39], Flan-T5 [7], LLaMA [46], LLaMA-2 [47] and Vicuna [6], have demonstrated impressive comprehension and generation capabilities. Researchers have leveraged the capabilities of LLMs to handle multimodal tasks, expanding them to multimodal large language models (MLLMs). For example, AnyGPT [53] employs LLaMA-2 [47] to construct an any-to-any multimodal language model. NExT-GPT [50] employs Vicuna [6] with multimodal adaptors and diffusion decoders to perform tasks across arbitrary combinations of text, images, videos, and audio. Recently, the works [21, 60] attempt to use LLMs for motion-related tasks. [60] uses LLaMA [46] to build a general-purpose motion generator, which, however, lacks the ability to comprehend motion. [21] leverages T5 to construct a unified motion-language model, but cannot deal with music modality.

---

[3]Note: in this paper, the term "audio" specifically refers to "music". This designation is adopted to avoid confusion between the initial letter "M" shared by both "music" and "motion," which could lead to ambiguity when these modalities are represented by their initials.

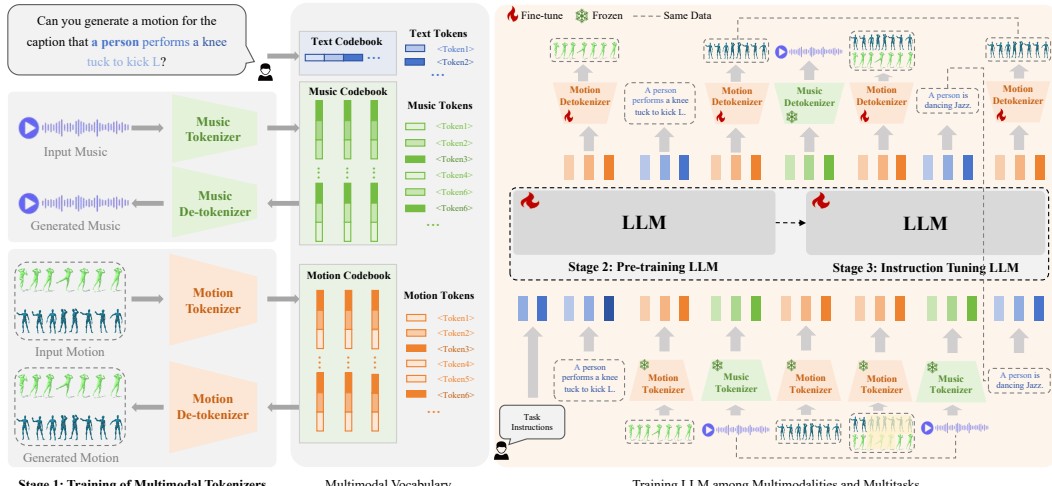

Figure 2: An overview of the M³GPT framework. M³GPT consists of multimodal tokenizers and a motion-aware language model. The training process of M³GPT consists of three stages: multimodal tokenizers training, modality-alignment pre-training, and instruction tuning.

## 3   Method

To enhance the comprehension and generation of motion-relevant modalities, we propose a unified multimodal framework named M³GPT. As illustrated in Fig. 2, M³GPT consists of multimodal tokenizers responsible for compressing raw motion and music data into discrete tokens (Sec. 3.1), and a motion-aware language model that learns to understand and generate motion tokens from LLMs by corresponding text and music (Sec. 3.2). To address motion-relevant tasks, we employ a three-stage training scheme encompassing multimodal tokenizers training, modality-alignment pre-training, and instruction tuning (Sec. 3.3). During the inference process, multimodal tokens are decoded back into their original representations by associated *de-tokenizers* (decoders of multimodal tokenizers), enabling various motion-relevant tasks to be executed via instructions (Sec. 3.4).

### 3.1   Multimodal tokenizers

As shown in Fig. 2, Multimodal tokenizers aim to discretize continuous human motion and music into language-like tokens, allowing the three modalities to be unified within a single language model.

**3D Human Motion Tokenizer.** To represent motion in discrete semantic tokens, we build a 3D human motion tokenizer based on Vector Quantized Variational Autoencoders (VQ-VAE) following [12, 62, 21, 60]. The motion tokenizer consists of a motion encoder $\mathcal{E}_m$ and a motion decoder $\mathcal{D}_m$, along with a codebook $\mathcal{B}_m = \left\{ b^1, b^2, \ldots, b^{N_m} \right\}$ containing $N_m$ discrete semantic vectors. Notably, to facilitate mutual enhancement between motion and dance data, we employ a shared tokenizer for both motions and dances, projecting them into a consistent and shared semantic space. Formally, given a 3D motion sequence $\boldsymbol{m} \in \mathbb{R}^{T_m \times d_m}$, where $T_m$ is the time length and $d_m$ is the dimensionality of each frame's pose, the motion encoder $\mathcal{E}_m$ that consists of several 1-D convolutional layers projects $\boldsymbol{m}$ to a latent embeddings $\boldsymbol{z} \in \mathbb{R}^{L_m \times d}$. Here, $L_m$ is the time interval after downsampling and $d$ is the latent dimension. Next, we transform $\boldsymbol{z}$ into a collection of codebook entries through discrete quantization. Specifically, the quantization process replaces each item of $\boldsymbol{z}$ with its nearest embedding in the codebook $\mathcal{B}_m$, obtaining the quantized latent vectors $\boldsymbol{e} \in \mathbb{R}^{L_m \times d}$ as follows:

$$\boldsymbol{e} = \underset{b^k \in \mathcal{B}_m}{\arg\min} \left\| \boldsymbol{z} - b^k \right\|_2 . \tag{1}$$

The motion decoder $\mathcal{D}_m$, which consists of several 1-D deconvolutional layers, projects the quantized embeddings back to raw motion space as $\hat{\boldsymbol{m}} = \mathcal{D}_m\left(\boldsymbol{e}\right)$. Following [21, 60], the motion tokenizer can be trained by the reconstruction loss, embedding loss and commitment loss as follows:

$$\mathcal{L}_{vq} = \|\hat{\boldsymbol{m}} - \boldsymbol{m}\|_1 + \|\text{sg}\left[\boldsymbol{z}\right] - \boldsymbol{e}\|_2^2 + \beta\|\boldsymbol{z} - \text{sg}\left[\boldsymbol{e}\right]\|_2^2 . \tag{2}$$

where $\text{sg}\left[\cdot\right]$ is the stop gradient, and $\beta$ is the factor that adjusts the weight of the commitment loss.

After training the motion tokenizer, a motion sequence $\boldsymbol{m}$ can be represented as a sequence of discrete codebook-indices of quantized embedding vector, namely *motion tokens* $\boldsymbol{q_m} \in \mathbb{R}^{L_m}$, as follows:

$$\boldsymbol{q_m} = \underset{k \in \{1,...,N_m\}}{\arg\min} \left\| \mathcal{E}_m\left(\boldsymbol{m}\right) - b^k \right\|_2 . \tag{3}$$

**Music Tokenizer.** For the music data, we adopt the VQ-VAE in Jukebox [9] as the music tokenizer, which consists of a music encoder $\mathcal{E}_a$, a music decoder $\mathcal{D}_a$ and a music codebook $\mathcal{B}_a$. Notably, the limited number of music samples in dance datasets makes it inadequate for training an effective music tokenizer. To leverage the strong representation ability of the tokenizer trained on the large-scale musical dataset, we use the pre-trained VQ-VAE from Jukebox [9], which has been trained on a dataset of 1.2 million songs. Specifically, we first segment each input music sample into 5-second music segments. Then, for each 5 seconds segment $\boldsymbol{a} \in \mathbb{R}^{T_a \times d_a}$, we use the pre-trained music tokenizer $\{\mathcal{E}_a, \mathcal{B}_a\}$ to encode $\boldsymbol{a}$ into a sequence of discrete codebook-indices $\boldsymbol{q_a} \in \mathbb{R}^{L_a}$ (namely *music tokens*) following Eq. 3.

### 3.2 Language Model Backbone

**Expanding Vocabulary.** To incorporate multimodal discrete representations into a pre-trained LLM, we expand the original text vocabulary $V_t$ in LLM with motion vocabulary $\mathcal{B}_m$ and music vocabulary $\mathcal{B}_a$, forming a new unified vocabulary $V = \{V_t, \mathcal{B}_m, \mathcal{B}_a\}$. To accommodate the expanded vocabulary, we extend the corresponding embedding and prediction layer of LLM, where the newly incorporated parameters are initialized randomly.

**Unified Multimodal Language Model.** Equipped with multimodal tokenizers, we can compress multimodal data into discrete token sequences. To be specific, employing the trained motion tokenizer and music tokenizer, the input motion $\boldsymbol{m} \in \mathbb{R}^{T_m \times d_m}$ and music $\boldsymbol{a} \in \mathbb{R}^{T_a \times d_a}$ can be mapped into a sequence of discrete motion tokens $\boldsymbol{q_m} \in \mathbb{R}^{L_m}$ and music tokens $\boldsymbol{q_a} \in \mathbb{R}^{L_a}$. Then equipped with a unified vocabulary $V$, we can formulate various motion-relevant tasks in a general format, where both input and output tokens come from the same vocabulary. These tokens can represent natural language, human motion, music, or any combination, depending on the specific task to be solved. This naturally enables the core LLM to unify motion comprehension and generation tasks in an autoregressive manner.

Following [21], we employ T5 [39] as the language model backbone, which is pre-trained on 750 GB of text tokens. By leveraging this pre-trained large language model, we can harness its powerful modeling capabilities and generalizability to develop a more user-friendly, motion-related human-computer interaction model.

### 3.3 Training Strategy

The training process is divided into three stages. The first stage is Multimodal Tokenizers Training, which focuses on learning the motion/music tokenizer to represent motion/music as discrete tokens. The second stage is Modality-Alignment Pre-training, which aims to align motion, music, and text modalities, and facilitate collaboration across different motion-relevant tasks. The third stage is Instruction Fine-Tuning, aimed at enhancing the model's instruction-following capability.

**Stage1: Multimodal Tokenizers Training.** We first train a motion tokenizer using the objective defined in Eq. 2. As for the music tokenizer, due to the limited music samples in existing dance datasets, we directly use the pre-trained VQ-VAE model from Jukebox [9]. This process allows any motion sequence and music to be represented as a sequence of tokens, enabling seamless integration with text within LLM. To ensure the stability of LLM training, the encoder of motion tokenizer and whole music tokenizer remain unchanged. Notably, we continue to optimize the decoder of motion tokenizer in subsequent training stages to further enhance the quality of generated motions.

**Stage2: Modality-Alignment Pre-training.** To enable LLM to handle discrete modalities, we utilize paired motion corpus to train LLM in a next-token prediction task. This process aims to align the text, music, and motion modalities for unified reasoning in LLM.

- **Joint optimization of LLM and motion de-tokenizer.** Human motion (especially dance) encompasses intricate details. Previous works [21, 60] keep the motion de-tokenizer fixed during training LLM, which hinders LLM's ability to perceive the distribution and details of motions. Specifically, in the output space of LLM, different motion tokens are treated as independent classes; therefore, the cost of classifying a motion token as semantic-similar token and semantic-distant

token is the same. Apparently, relying solely on LLM's autoregressive loss is insufficient for capturing the details of motion. To address this problem, we jointly optimize LLM and motion de-tokenizer in stage2 and stage3. This strategy enables the reconstruction error signals in raw motion space to backpropagate to LLM, enhancing LLM's ability to generate the details of motion. With the goal of minimizing L1 loss between the predicted and real motion, we search for the motion's token sequence that could minimize this L1 loss in original motion space. As the motion de-tokenizer continuously optimizes, the target motion's token sequence, which supervises LLM training, dynamically changes. This dynamic adjustment reduces L1 loss progressively, achieving joint optimization.

- **Synergy learning of multitasks.** Although aligning text with one additional modality is relatively straightforward, integrating multiple modalities (*e.g.*, motion, text, and music) within a single framework poses a significant challenge. Additionally, as noted in [4], multitask joint training usually achieves inferior performance on each individual task compared to single-task training. This phenomenon is also observed in our *text-to-motion* task, as shown in Tab. 2. We argue that the large modality difference among different motion-relevant tasks (*e.g.*, *music-to-dance* and *text-to-motion*) prevents the model from effectively establishing connections between these tasks. Thus it is difficult for the model to identify a common optimization direction that benefits all tasks.

  As 'text' serves as a highly semantic descriptor for other modalities and is the most familiar and well-modeled modality to LLM, we use 'text' as a bridge to align motion, text, and music data, thereby mitigating conflicts in aligning multiple modalities. Initially, we construct paired textual descriptions for music samples in the dance datasets. Specifically, we use the style annotations of the music to create paired texts, such as 'a person is dancing Jazz'. Then, we construct two auxiliary tasks using the generated pairs of music and text, *i.e.*, *music-to-text* and *text-to-dance*. Through these two auxiliary tasks, M³GPT implicitly learns to decompose the complex *music-to-dance* task into two simpler tasks *music-to-text* and *text-to-dance*. Additionally, with a shared tokenizer for motion and dance, *text-to-dance* and *text-to-motion* tasks occupy the same matching space, and thus can mutually reinforce each other. In this way, M³GPT builds the synergies between the two primary motion generation tasks, music-to-dance and text-to-motion, facilitating mutual reinforcement, as shown in Tab. 2.

Combining the above analysis, we jointly train LLM and motion de-tokenizer using a mixture of motion comprehension and generation tasks, along with two auxiliary *music-to-text* and *text-to-dance* tasks. Besides the auxiliary tasks, we consider 2 basic motion comprehension tasks, *i.e.*, motion-to-text and dance-to-music, and 4 basic motion generation tasks, *i.e.*, text-to-motion, music-to-dance, motion prediction and motion in-between. Formally, for a specific task, we denote the source input consisting of a sequence of tokens as $\boldsymbol{q}_s = \left\{\boldsymbol{q}_s^i\right\}_{i=1}^{L_s}$, the target output as $\boldsymbol{q}_t = \left\{\boldsymbol{q}_t^i\right\}_{i=1}^{L_t}$, LLM predicts the probability distribution of potential next token at each step $p_\theta\left(\boldsymbol{q}_t^i|\boldsymbol{q}_t^{<i}, \boldsymbol{q}_s\right)$ in an autoregressive manner. For motion generation tasks, we add a reconstruction loss. Specifically, when the output tokens are motion tokens, we pass them to motion de-tokenizer to generate a motion sequence (denoted as $\hat{\boldsymbol{m}}$), where a reconstruction loss is then employed for guidance. Overall, during this training process, the objective is to maximize the log-likelihood of the data distribution and minimize the reconstruction error within raw motion space:

$$\mathcal{L} = \sum_{i=0}^{L_t-1} \log p_\theta\left(\boldsymbol{q}_t^i|\boldsymbol{q}_t^{<i}, \boldsymbol{q}_s\right) + \lambda\left\|\hat{\boldsymbol{m}} - \boldsymbol{m}\right\|_1, \tag{4}$$

where $\boldsymbol{m}$ denotes the ground-truth for $\hat{\boldsymbol{m}}$ generated by motion de-tokenizer, and $\lambda$ is a hyper-parameter to adjust the weight for reconstruction loss.

**Stage3: Instruction Fine-Tuning.** To enhance the generalization and instruction-following capability of M³GPT, we construct a multimodal instruction dataset with resort to GPT-4, building upon existing motion datasets. Specifically, we define 11 core tasks, each comprising 200/50/50 training/validation/test instruction templates. For example, an instruction for text-to-motion task could be "Create a motion that complements the poetic elements in <Caption_Placeholder>", with <Caption_Placeholder> standing for any text sequence; an instruction for music-to-dance could be "Script a dance that adapts to the tempo shifts in <Audio_Placeholder>", with <Audio_Placeholder> standing for any music sequence. Further details are available in Appendix B.4.

### 3.4 Inference M$^3$GPT

During inference, we evaluate M$^3$GPT's performance across multiple motion-relevant tasks and datasets (Sec. 4.2 and Appendix (C, D). Also, we consider two challenging dance generation tasks to evaluate the zero-shot generalization ability of M$^3$GPT:

**(1) Generating long-duration dances from long music.** Long-duration dance generation involves creating uninterrupted, coherent dance sequences based on a single piece of music. Due to the limitation of computational cost and memory overload, we train M$^3$GPT on the task of 5-second music-to-dance generation. Conversely, during inference, we can combine this training task *music-to-dance* to generate an initial 5-second dance segment, and an unseen zero-shot task *music+dance-to-dance* that recursively generates subsequent dance segments conditioned on both music and previously generated dance segments, to perform long-duration and coherent dance generation.

**(2) Generating dance controlled by both music and text.** Integrating music and text as control signals in dance generation (*music+text-to-dance*) augments music-to-dance task with text modality. This process guides the generated dances to synchronize with particular actions described in input texts. Thanks to the tokenizer mechanism, M$^3$GPT can seamlessly combine music and text in LLM's input, enabling the integration of text instructions to produce a wide variety of dance sequences.

## 4 Experiments

### 4.1 Experimental setup

**Datasets and Preprocessing.** We use a large-scale text-to-motion dataset: Motion-X [29], and two music-to-dance datasets: AIST++ [24] and FineDance [25]. Notably, the 3D pose annotations differ among these datasets, therefore, we standardize their processing for uniform usage. Specifically, we select 22 joints common to these datasets and preprocess the motion samples following [13], resulting in motion sequences with identical representation. Further details on datasets and preprocessing are provided in Appendix B.1.

**Evaluation Metrics.** Different tasks employ distinct evaluation metrics. We use the most common evaluation metrics to assess the performance of M$^3$GPT for each task. **(1) Text-to-Motion**. Following [21, 12], we use *Frechet Inception Distance (FID)*, *Diversity (Div)*, *R-Precition* that calculates the top-1 motion-to-text retrieval accuracy (R TOP1). **(2) Motion-to-Text**. Following [21], we use linguistic metrics like *BLEU*, *CIDEr*, along with *R-Precision* for evaluating motion-to-text alignment. **(3) Music-to-Dance**. Following [26, 52], we use *FID*, *Diversity* and *Beat Align Score (BAS)* on kinetic features [22] (denoted as "$k$") to evaluate the dance generation quality. Notably, as noted in [48], the geometric features [33] are unreliable as a measure of dance generation quality. So we only use the kinetic features for evaluation. **(4) Dance-to-Music**. Following [52], we use *Beats Coverage Scores (BCS)*, *Beats Hit Scores (BHS)*, and *F1* score to evaluate music generation quality. **(5) Motion Prediction and In-Between**. Following [21], we use *FID* and *Diversity* to measure the consistency between the provided pose conditions and generated motion. More details and results on other evaluation metrics are provided in Appendix B.2 and C.

**Implementation Details.** For motion tokenizer, we set the codebook size to $512$. As for music tokenizer, we use the pre-trained VQ-VAE from Jukebox with a codebook size of $2048$. In term of temporal downsampling rate, the motion encoder uses a rate of $4$, while the music encoder uses a rate of $128$. We utilize T5 base [39] as our language model backbone. For training the motion tokenizer, we use Adam as the optimizer with a batch size of $1000$ and an initial learning rate of $10^{-4}$. To train the language model backbone, we employ the Adafactor_dev optimizer and use CosineAnnealingLR as the learning rate scheduler. The learning rate is set to $2 \times 10^{-4}$ for pre-training stage, and $10^{-4}$ for instruction fine-tuning stage. For hyperparameter settings, $\lambda$ in Eq. 4 is set to $0.2$, and $\beta$ in Eq. 2 is set to $0.02$. All our experiments are conducted on 8 NVIDIA A40 GPUs. To evaluate the model's performance across different platforms, we also test our trained M$^3$GPT with T5-base on Ascend 910B NPUs. Further details on implementation and hyperparameter analysis are provided in Appendix E.

Table 2: Evaluation of synergy learning and joint optimization of LLM and motion de-tokenizer on Text-to-Motion (Motion-X [29]) and Music-to-Dance (AIST++ [24]). T2M: Text-to-Motion. A2D: Music-to-Dance. T2D: Text-to-Dance. A2T: Music-to-Text. *Trained single task* refers to a model trained and tested on a single task. *Pre-trained* and *Instruction-tuned* indicate the model after pre-training (stage2) and instruction tuning (stage3), followed by direct testing on each task. The arrows (↑) indicate that higher values are better. The arrows (↓) indicate that smaller values are better. **Bold** indicates the best result.

| Methods | Re-Optimizing motion de-tokenizer | Text-to-Motion | | | Music-to-Dance | | |
|---|---|---|---|---|---|---|---|
| | | R TOP1 ↑ | FID ↓ | Div↑ | $FID_k$ ↓ | $Div_k$ ↑ | BAS ↑ |
| Ground Truth | - | 0.675 | 0.009 | 2.316 | 17.10 | 8.19 | 0.2374 |
| Trained single task | | 0.645 | 0.081 | 2.124 | 83.33 | 5.18 | 0.1835 |
| Trained single task | ✓ | **0.656** | **0.078** | 2.133 | 75.47 | 5.57 | 0.1884 |
| T2M+A2D | | 0.564 | 0.094 | 2.080 | 51.26 | 6.73 | 0.2037 |
| T2M+A2D | ✓ | 0.578 | 0.092 | 2.106 | 47.71 | 7.47 | 0.1958 |
| T2M+A2D+T2D+A2T | | 0.617 | 0.093 | 2.110 | 42.70 | 7.54 | 0.2084 |
| T2M+A2D+T2D+A2T | ✓ | 0.626 | 0.088 | 2.197 | 25.24 | **7.63** | **0.2217** |
| M³GPT (Pretrained without T2D and A2T) | | 0.526 | 0.105 | 2.058 | 40.71 | 7.47 | 0.2030 |
| M³GPT (Pretrained without T2D and A2T) | ✓ | 0.547 | 0.104 | 2.099 | 37.14 | 7.61 | 0.2005 |
| M³GPT (Pre-trained) | | 0.598 | 0.089 | 2.218 | 32.71 | 7.43 | 0.2090 |
| M³GPT (Pre-trained) | ✓ | 0.601 | 0.092 | 2.251 | 27.65 | 7.52 | 0.2105 |
| M³GPT (Instruction-tuned) | | 0.606 | 0.091 | 2.251 | 28.46 | 7.49 | 0.2052 |
| M³GPT (Instruction-tuned) | ✓ | 0.615 | 0.093 | **2.253** | **24.34** | 7.50 | 0.2056 |

## 4.2 Ablation Studies

In this section, we conduct ablation studies to validate the effectiveness of our method. We use the same model architecture throughout the experiments. The ablation results are shown in Tab. 2.

**Effectiveness of joint optimization of LLM and motion de-tokenizer.** Different from previous works [21, 60] that fix motion de-tokenizer during training LLM, we jointly optimize LLM and motion de-tokenizer in stage2 and stage3, as detailed in Sec. 3.3. As shown in Tab. 2, the joint optimization consistently brings performance gains across various metrics and most settings. Specifically, it largely enhances the fidelity of generated dances, reflected in a notable decrease in $FID_k$ score. We also notice a minor increase (less than 0.003) in FID of text-to-motion task in M³GPT. The possible reason is that the motion patterns controlled by text are relatively simple, making LLM optimized solely in discrete semantic space adequate for text-to-motion. Conversely, dances involve greater complexity, necessitating the joint optimization of motion decoder to accurately capture intricate dance movements without compromising information.

**Effectiveness of synergy learning of multitasks.** During the training of M³GPT, we introduce a synergy multitask learning strategy by constructing two auxiliary tasks: Text-to-Dance (T2D) and Music-to-Text (A2T), as detailed in Sec. 3.3. As shown in Tab. 2, the inclusion of T2D and A2T consistently brings performance gains across various metrics on both text-to-motion and music-to-dance tasks. Specifically, for music-to-dance, the $FID_k$ score is decreased by nearly 10 points, indicating that the synergy learning helps generate more realistic dances. We argue that by incorporating these two auxlary tasks, M³GPT implicitly learns to decompose the complex music-to-dance into two simpler tasks, music-to-text and text-to-dance. This way, the text-to-motion task can assist in learning the music-to-dance task, thereby enhancing its performance.

## 4.3 Comparisons with State-of-the-arts

In this section, we compare our M³GPT with state-of-the-arts on multiple core motion-relevant tasks. We respectively report the comparison results on text-to-motion dataset, Motion-X [29], and music-to-dance datasets, AIST++ [24] and FineDance [25]. More quantitative and qualitative results are provided in Appendix C and D. Also, in the supplementary material's zip file, we provide the render videos of generated motions/dances and generated music files by our M³GPT.

**Main results on text-to-motion dataset.** On the text-to-motion dataset, Motion-X, we evaluate M³GPT on 4 tasks, *i.e.*, text-to-motion, motion-to-text, motion prediction, and motion in-between. The comparison results are shown in Tab. 3. As shown, M³GPT achieves competitive performance across all evaluated tasks, highlighting its capability to address diverse motion tasks in a single model. Also, for text-to-motion task, *M³GPT (instruction-tuned only T2M)*, which combines multitask pre-training and instruction fine-tuning solely on T2M task, yields better performance than *Trained single*

Table 3: Comparison results on Motion-X [29] dataset. The evaluation metrics are computed using the encoder introduced in Appendix A. Empty columns of previous methods indicate that they can not handle the task. *Instruction-tuned only T2M* indicates the model that is initially pre-trained on multiple tasks, followed by instruction tuning solely on text-to-motion task.

| Methods | Text-to-Motion | | | Motion-to-Text | | | Motion Prediction | | Motion In-between | |
|---|---|---|---|---|---|---|---|---|---|---|
| | R TOP1↑ | FID↓ | Div↑ | R TOP3↑ | Bleu@4↑ | CIDEr↑ | FID↓ | Div↑ | FID↓ | Div↑ |
| Real | $0.675^{\pm0.003}$ | $0.009^{\pm0.000}$ | $2.316^{\pm0.011}$ | 0.881 | - | - | 0.009 | 2.316 | 0.009 | 2.316 |
| MLD [5] | $0.612^{\pm0.003}$ | $0.122^{\pm0.008}$ | $2.267^{\pm0.018}$ | - | - | - | - | - | - | - |
| T2M-GPT [55] | $0.647^{\pm0.002}$ | $0.101^{\pm0.005}$ | $2.270^{\pm0.033}$ | - | - | - | 0.814 | 1.755 | - | - |
| MotionDiffuse [57] | $0.659^{\pm0.002}$ | $0.075^{\pm0.004}$ | $2.220^{\pm0.022}$ | - | - | - | - | - | - | - |
| TM2T [15] | $0.581^{\pm0.003}$ | $0.148^{\pm0.003}$ | $2.005^{\pm0.034}$ | 0.806 | **12.13** | 20.16 | - | - | - | - |
| MDM [44] | $0.472^{\pm0.008}$ | $0.078^{\pm0.000}$ | $2.133^{\pm0.012}$ | - | - | - | 1.028 | 1.746 | 0.831 | 1.768 |
| MoMask [16] | $\mathbf{0.668}^{\pm0.003}$ | $\mathbf{0.074}^{\pm0.004}$ | $2.241^{\pm0.016}$ | - | - | - | - | - | 0.626 | 1.884 |
| MotionLCM [8] | $0.658^{\pm0.005}$ | $0.078^{\pm0.003}$ | $2.206^{\pm0.026}$ | - | - | - | - | - | | |
| MotionGPT[21] | $0.659^{\pm0.003}$ | $0.078^{\pm0.001}$ | $2.166^{\pm0.026}$ | 0.840 | 11.21 | 31.36 | 0.701 | 1.818 | 0.648 | 1.875 |
| Trained single task | $0.656^{\pm0.002}$ | $0.078^{\pm0.002}$ | $2.133^{\pm0.012}$ | 0.767 | 10.14 | 22.92 | 0.774 | 1.778 | 0.692 | 1.810 |
| M³GPT (Pre-trained) | $0.601^{\pm0.002}$ | $0.092^{\pm0.002}$ | $2.251^{\pm0.012}$ | 0.834 | 11.00 | 24.12 | 0.707 | **1.874** | **0.604** | 1.879 |
| M³GPT (Instruction-tuned) | $0.615^{\pm0.003}$ | $0.093^{\pm0.002}$ | $2.253^{\pm0.026}$ | **0.845** | 11.50 | **42.93** | 0.682 | 1.838 | 0.612 | **1.900** |
| M³GPT (Instruction-tuned only T2M) | $0.661^{\pm0.003}$ | $0.076^{\pm0.002}$ | $\mathbf{2.273}^{\pm0.026}$ | - | - | - | - | - | - | - |

Table 4: Comparison results on Motion-X [29] dataset based on Ascend 910B NPUs.

| Methods | Text-to-Motion | | | Motion-to-Text | | | Motion Prediction | | Motion In-between | |
|---|---|---|---|---|---|---|---|---|---|---|
| | R TOP1↑ | FID↓ | Div↑ | R TOP3↑ | Bleu@4↑ | CIDEr↑ | FID↓ | Div↑ | FID↓ | Div↑ |
| Trained single task | $0.654^{\pm0.002}$ | $0.081^{\pm0.003}$ | $2.304^{\pm0.017}$ | 0.763 | 10.16 | 22.89 | 0.776 | 1.818 | 0.712 | 1.880 |
| M³GPT (Pre-trained) | $0.596^{\pm0.002}$ | $0.096^{\pm0.003}$ | $2.241^{\pm0.018}$ | 0.831 | 11.05 | 24.03 | 0.710 | **1.882** | **0.608** | 1.874 |
| M³GPT (Instruction-tuned) | $0.612^{\pm0.002}$ | $0.094^{\pm0.002}$ | $2.276^{\pm0.021}$ | **0.846** | **11.52** | **42.64** | 0.684 | 1.841 | 0.621 | **1.903** |
| M³GPT (Instruction-tuned only T2M) | $\mathbf{0.659}^{\pm0.003}$ | $\mathbf{0.078}^{\pm0.003}$ | $\mathbf{2.314}^{\pm0.023}$ | - | - | - | - | - | - | - |

Table 5: Comparison results on AIST++ [24] and FineDance [25] datasets.

| Methods | Music-to-Dance on AIST++ | | | Music-to-Dance on FineDance | | | Dance-to-Music on AIST++ | | |
|---|---|---|---|---|---|---|---|---|---|
| | $\text{FID}_k\downarrow$ | $\text{Div}_k\uparrow$ | BAS↑ | $\text{FID}_k\downarrow$ | $\text{Div}_k\uparrow$ | BAS↑ | BCS↑ | BHS↑ | F1↑ |
| Real | 17.10 | 10.60 | 0.2374 | - | - | 0.2120 | - | - | - |
| FACT [24] | 35.35 | 5.94 | 0.2209 | 113.38 | 3.36 | 0.1831 | - | - | - |
| Bailando [42] | 28.16 | 7.83 | 0.2332 | 82.81 | 7.74 | 0.2029 | - | - | - |
| EDGE [48] | 42.16 | 3.96 | 0.2334 | 94.34 | 8.13 | 0.2116 | - | - | - |
| Lodge [26] | 37.09 | 5.58 | **0.2423** | 45.56 | 6.75 | **0.2397** | - | - | - |
| Foley [11] | - | - | - | - | - | - | 96.4 | 41.0 | 57.5 |
| CMT [10] | - | - | - | - | - | - | 97.1 | 46.2 | 62.6 |
| D2MGAN [64] | - | - | - | - | - | - | 95.6 | 88.7 | 93.1 |
| CDCD [65] | - | - | - | - | - | - | 96.5 | 89.3 | 92.7 |
| LORIS [52] | - | - | - | - | - | - | **98.6** | 90.8 | 94.5 |
| Trained single task | 75.47 | 5.57 | 0.1884 | 128.37 | 6.48 | 0.2036 | 93.9 | 93.6 | 92.8 |
| M³GPT (Pre-trained) | 27.65 | 7.52 | 0.2105 | 92.35 | 7.67 | 0.2134 | 93.4 | 93.8 | 94.2 |
| M³GPT (Instruction-tuned) | 24.34 | 7.50 | 0.2056 | 86.47 | 7.75 | 0.2158 | 93.6 | **94.0** | 94.9 |
| M³GPT (Instruction-tuned for single task) | **23.01** | **7.85** | 0.2261 | **42.66** | **8.24** | 0.2231 | 94.3 | **94.0** | **95.0** |

Table 6: Comparison results on AIST++ [24] and FineDance [25] datasets based on Ascend 910B NPUs.

| Methods | Music-to-Dance on AIST++ | | | Music-to-Dance on FineDance | | | Dance-to-Music on AIST++ | | |
|---|---|---|---|---|---|---|---|---|---|
| | $\text{FID}_k\downarrow$ | $\text{Div}_k\uparrow$ | BAS↑ | $\text{FID}_k\downarrow$ | $\text{Div}_k\uparrow$ | BAS↑ | BCS↑ | BHS↑ | F1↑ |
| Trained single task | 77.32 | 5.61 | 0.1860 | 134.66 | 6.52 | 0.2088 | 93.8 | 93.6 | 92.7 |
| M³GPT (Pre-trained) | 27.99 | 7.61 | 0.2102 | 91.39 | 7.71 | 0.2128 | 93.4 | 93.6 | 94.2 |
| M³GPT (Instruction-tuned) | 25.05 | 7.48 | 0.2072 | 88.25 | 7.76 | 0.2160 | 93.5 | 93.8 | 94.7 |
| M³GPT (Instruction-tuned for single task) | **23.68** | **7.83** | 0.2264 | **43.78** | **8.39** | 0.2225 | 94.3 | 94.0 | 94.9 |

*task* that only trains the model on T2M task. Tab. 4 presents the results of testing on the Ascend 910B NPUs, where M³GPT also achieves comparably good performance. These results demonstrate that multitask pre-training can enhance the performance of individual tasks across different computation platforms. Further results of M³GPT trained on NPUs will be presented later.

**Main results on music-to-dance datasets.** On the music-to-dance datasets, AIST++ and FineDance, we evaluate M³GPT on 2 tasks, *i.e.*, music-to-dance and dance-to-music. As shown in Tab. 5, in general, the performance of multitask pre-training and instruction fine-tuning in M³GPT outperforms single-task training, underscoring the effectiveness of multitask training for dance-relevant tasks. Also, M³GPT achieves competitive performance on most metrics. For music-to-dance, the best-performing

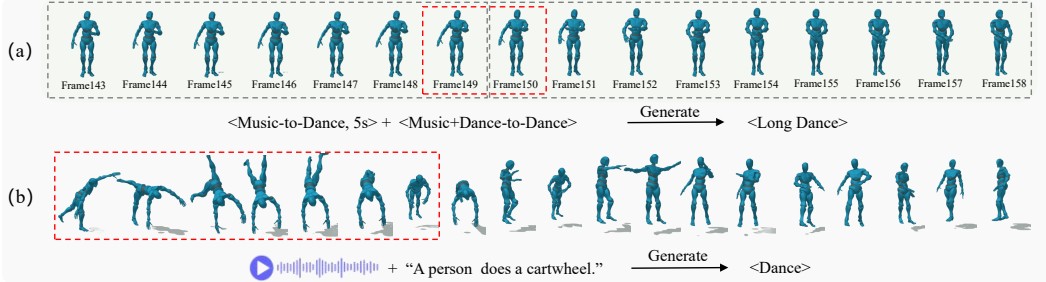

Figure 3: Qualitative results for long-term dance and music-text conditioned dance generation of M$^3$GPT.

method on the FineDance dataset is Lodge [26]. This approach features a specialized two-stage architecture for generating long-duration dance sequences, progressively refining movements from coarse to fine granularity using a diffusion model. On AIST++ dataset, M$^3$GPT reports the best FID$_k$ of $24.34$ for music-to-dance task, the best BHS and F1 of $94.0$ and $94.9$ for dance-to-music task. The results in Tab. 6, tested on the Ascend 910B NPUs, also demonstrate that multitask training can enhance the performance of both music-to-dance and dance-to-music tasks.

### 4.4 Evaluation on Zero-Shot Tasks

In this section, we explore M$^3$GPT's capabilities in handling zero-shot tasks, as mentioned in Sec. 3.4. Fig. 3 (a) shows the long-term dance generation. As seen, M$^3$GPT can generate a coherent dance sequence by seamlessly integrating the music-to-dance and zero-shot *music+dance-to-dance* tasks. Fig. 3 (b) shows the generated 3D dance with both music and text instruction. We can see that M$^3$GPT maintains plausible visual results in accordance with input text instructions (*cartweel*), underscoring its remarkable zero-shot generalization capability.

## 5 Conclusion

In this paper, we present M$^3$GPT, a unified framework for comprehending and generating motion aligned with both text and music modalities. By employing text as a bridge, we build connections and synergies between different motion-relevant tasks, facilitating mutual reinforcement. Additionally, the joint optimization of LLM and motion de-tokenizer further enriches the details of generated motion, enhancing overall motion generation quality. Our extensive evaluations across various motion-relevant tasks demonstrate the effectiveness of M$^3$GPT in both motion comprehension and generation. Besides, M$^3$GPT exhibits strong zero-shot generalization abilities, enabling it to handle previously unseen and challenging motion-relevant tasks.

**Limitations and Broader Impacts.** Although our M$^3$GPT has successfully processed signals from motion, text, and music modalities for motion comprehension and generation, it focuses on modeling human body movements, excluding hands and faces details. Future research can extend the scope of M$^3$GPT to include hands and facial modeling.

**Acknowledgments.** This work is sponsored by the National Natural Science Foundation of China (NSFC) under Grant 62306301, the National Postdoctoral Program for Innovative Talents under Grant BX20220310, and the CAAI-CANN Open Fund developed on the OpenI Community. It is also supported by the project of Peng Cheng Laboratory (PCL2023A08).

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

# A  Text-Motion Alignment Model

Due to the lack of a powerful and publicly available text-motion alignment model, we independently leverage existing datasets to develop a functional text-motion alignment model, which is used to evaluate tasks such as text-to-motion, motion-to-text, motion prediction, and motion in-between. We adopt the motion encoder architecture $E_m$ from [14] and use the pretrained CLIP [38] ViT-B/32 model for the text encoder $E_t$. As depicted in Fig. 4, the training of the text-motion alignment model is split into two phases: pre-training the motion auto-encoder and text-motion contrastive learning. During the pre-training phase, we use motion data from the text-to-motion, and dance data from the music-to-dance tasks. This stage employs a reconstruction loss to ensure the model achieves a robust initial state capable of extracting an expressive motion representation. In the text-motion contrastive learning phase, we utilize text-motion pair data from the text-to-motion task. To train a more robust text-motion alignment model, we employ text-motion training dataset data along with motion data reconstructed by Motion VQ-VAE for text-motion contrastive training. We incorporate an adapter MLP layer into both the motion and text encoders to align the dimensions of $\mathbf{z}_m$ and $\mathbf{z}_t$ at 512. This setup facilitates the alignment of text and motion in the representational space. The motion reconstruction loss $L_{\text{recon\_motion}}$ for the pre-training stage and the contrastive learning loss $L_{\text{infoNCE}}$ for the aligning stage are used to optimize this model, as follows,

$$\mathcal{L}_{\text{recon\_motion}} = \|x - \tilde{x}\|^2 \tag{5}$$

$$\mathcal{L}_{\text{infoNCE}} = -\frac{1}{N} \sum_{i=1}^{K} \log \left( \frac{\exp(\langle z_i', z_i^t \rangle / \tau)}{\sum_{j=1}^{K} \exp(\langle z_i', z_j^t \rangle / \tau)} \right) \tag{6}$$

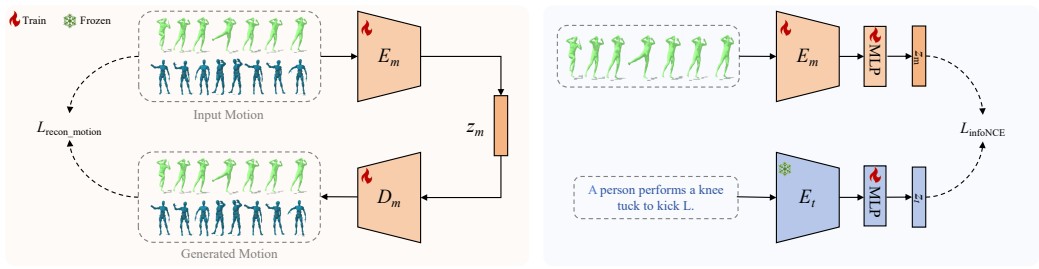

Figure 4: Pipeline of Text-Motion Alignment Model. The training of the text-motion alignment model includes two stages: pre-training motion auto-encoder and text-motion contrastive learning.

# B  Details for Training and Evaluating

## B.1  Data Introductions and Preprocessing

We leverage the largest available text-to-motion dataset, Motion-X [29], along with widely-used music-to-dance datasets, AIST++ [24] and FineDance [25], for our multitask training regimen. Motion-X is used for text-to-motion, motion-to-text, motion prediction, and motion in-between tasks, while AIST++ and FineDance datasets support both music-to-dance and dance-to-music tasks, and are also adapted for motion prediction and in-between tasks to enhance our training resources.

Motion-X includes 15.6 million precise 3D whole-body SMPL-X [36] pose annotations across 81.1K motion sequences with sequence-level semantic text descriptions. AIST++ contains 1,409 dance motion sequences across 10 dance genres with SMPL [30] pose annotations, and FineDance provides 7.7 hours of dance, totaling 831,600 frames with SMPL-H [41] poses at 30 fps across 16 dance genres. Tab. 7 shows the training datasets and their corresponding sample numbers that we use to train our model.

We standardize data across these datasets by selecting the 22 common joints and normalizing each motion sequence to face the same direction and to run at 30 fps. We use a processing technique consistent with prior research [14, 5, 21] that integrates joint velocities, positions, and rotations for consistent motion representation, facilitating effective utilization across tasks. For the music data, we use the Librosa toolkit [32] to load raw .wav data at a sampling rate of 22050 Hz, processed into features by the Jukebox encoder [9]. To optimize the use of these datasets, we strategically employ data from specific datasets for different tasks. During training for music and dance tasks, we randomly select 5-second segments from complete music tracks and corresponding dance

segments as training samples, setting the sample length to 6.25 seconds for motion prediction and in-between tasks with AIST++ and FineDance. When assessing music-to-dance on the FineDance dataset, we don't evaluate all 5-second samples directly. Instead, we generate continuous long dance sequences using music-to-dance and motion prediction, then segment these into 30-second samples for evaluation to align with Lodge's testing methodology.

Table 7: The training datasets and sample numbers for different tasks.

| Tasks | T2M | M2T | Motion Prediction/In-between | A2D | D2A | A2T | T2D |
|---|---|---|---|---|---|---|---|
| Training dataset | Motion-X | Motion-X | Motion-X/AIST++/FineDance | AIST++/FineDance | AIST++/FineDance | AIST++/FineDance | AIST++/FineDance |
| Training samples number | 64867 | 64867 | 64867/952/177 | 952/177 | 952/177 | 952/177 | 952/177 |

## B.2 Comprehensive Evaluation Metrics

Different tasks utilize specific evaluation metrics. We use the consistent evaluation metrics following prior research [14, 21, 12, 26, 64, 52].

- Text-to-Motion. We measure the discrepancy between generated and actual motion features using Frechet Inception Distance (FID), assess diversity (Div) among all generated motion sequences, and evaluate motion-text semantic correlation with R-precision. Multimodal Distance (MM Dist) quantifies the disparity between motions and texts. A specialized model developed for evaluating the text-to-motion task on the Motion-X dataset with 22 joints is detailed in Appendix A.

- Motion-to-Text. We use linguistic metrics including BLEU [35], ROUGE [28], CIDEr [49], and BertScore [59], along with R-Precision and MM Dist to assess alignment between generated text and motion.

- Music-to-Dance. We employ the evaluation framework recommended by FACT [24] and Bailando [42], utilizing FID, Diversity, and Beat Align Score (BAS) to gauge dance quality. In our paper, we use kinetic features to compute FID and Diversity.

- Dance-to-Music. We use metrics from [52, 64] such as Beats Coverage Scores (BCS), Beats Hit Scores (BHS), F1 scores, Beats Coverage Std (CSD), and Beats Hit Std (HSD) to evaluate music generation quality.

- Motion Prediction and Motion In-between. We use Average Displacement Error (ADE) and Final Displacement Error (FDE) to assess the quality of predicted motion. The text-motion alignment model aids in evaluating motion prediction performance.

## B.3 Distributed Training for Multitasks

We employ a single-node multi-GPU distributed training strategy for M³GPT, distributing each task across separate GPUs to facilitate multitask training through shared model parameters. This method allows us to tailor the maximum token length of the LLM for each task, based on the longest sample sequence typical for that task. Specifically, we set the maximum LLM token length to 192 for tasks such as text-to-motion, motion-to-text, motion prediction, and motion in-between. For tasks involving the music modality, such as music-to-dance and dance-to-music, the maximum length is set at 980. This task-specific configuration enables us to optimize batch sizes effectively, thus maximizing GPU utilization.

In our experimental setup, the batch size is set to 40 for the text-to-motion, text-to-dance, and motion-to-text tasks, and 4 for the music-to-dance, dance-to-music, and music-to-text tasks. For motion prediction and motion in-between tasks, the batch size is set to 45. We also establish the number of iterations for pre-training at 1,000,000, instruction fine-tuning at 200,000. This structured approach ensures that each task is optimally processed, enhancing the efficiency and effectiveness of our training regimen.

## B.4 Tasks for Pre-training and Instruction tuning

As shown in Fig. 5, we define 11 core motion-related tasks for the instruction tuning of M³GPT, including text-to-motion, random-to-motion, motion-to-text, random-to-text, motion prediction, motion in-between, music-to-dance, dance-to-music, random-to-music, text-to-dance, and music-to-text. The tasks of text-to-dance and music-to-text were specifically constructed based on the music-to-dance datasets. Random-to-X represents the unconstrained generation of motion, text and music. In Tab. 8, we present a selection of command templates for each task, where <Motion_Placeholder>, <Caption_Placeholder>, and <Music_Placeholder> respectively represent the motion sequence (including dance sequence), textual description, and music segment from the training data.

## C Quantitative Results and Comparisons with SOTA Methods

In this section, we compare the performance of our M³GPT with existing SOTA methods on a broader set of metrics for each task across three datasets: Motion-X [29], AIST++ [24], and FineDance [25]. Tab. 9 shows

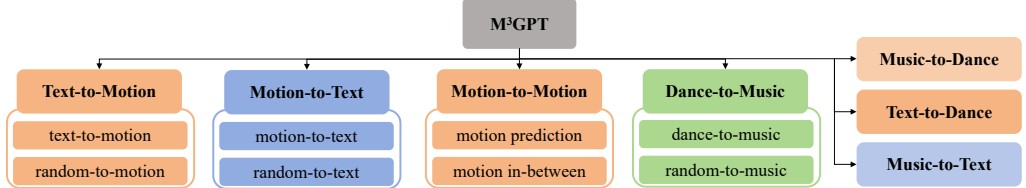

Figure 5: Tasks for M³GPT pre-training and instruction tuning. Random represents the unconstrained generation of motion/text/music in the corresponding task.

Table 8: Examples of instruction templates for each task when instruction tuning M³GPT.

| Task | Instruction template | Output |
|------|---------------------|--------|
| text-to-motion | Design a motion that illustrates the emotions conveyed by <Caption_Placeholder>. How could you express the resilience mentioned in <Caption_Placeholder> through motion? Can you develop a motion that captures the existential debate in <Caption_Placeholder>? | <Motion_Placeholder> |
| random-to-motion | Can you generate a motion randomly? Please generate a random motion. Display a motion for me. | <Motion_Placeholder> |
| motion-to-text | What themes are explored through the motion in <Motion_Placeholder>? Can you describe the motion <Motion_Placeholder> with texts? How would you interpret the actions depicted in <Motion_Placeholder>? | <Caption_Placeholder> |
| random-to-text | Can you generate a text description for motion randomly? Give me a caption which describes a action. How can we describe a motion with texts? | <Caption_Placeholder> |
| motion prediction | What movements would suitably follow the thematic climax of <Motion_Placeholder>? What steps might deepen the emotional expression seen in <Motion_Placeholder>? What movements could follow to resolve the suspense built in <Motion_Placeholder>? | <Motion_Placeholder> |
| motion in-between | What new character dynamics could the middle section of <Motion_Placeholder> explore? Infer the type of dramatic climax that the masked section of <Motion_Placeholder> might contain. What potential themes of ascent or descent could be explored in the middle of <Motion_Placeholder>? | <Motion_Placeholder> |
| music-to-dance | Script a dance that adapts to the tempo shifts in <Music_Placeholder>. Create a dance that would visually mimic the lyrical journey in <Music_Placeholder>. Compose a dance that explores the genre characteristics of <Music_Placeholder>. | <Motion_Placeholder> |
| dance-to-music | Can you design a music for this dance <Motion_Placeholder>? Please create a music based on this dance <Motion_Placeholder>. Arrange a symphony that captures the shifts in <Motion_Placeholder>. | <Music_Placeholder> |
| random-to-music | Please generate a music for a dance randomly. Can you generate a music with dance style? Creat a music for any style dance. | <Music_Placeholder> |
| text-to-dance | Please generate a dance based on the caption <Caption_Placeholder>. Create a dance for the text <Caption_Placeholder>. Generate a dance that corresponds to the textual description <Caption_Placeholder>. | <Motion_Placeholder> |
| music-to-text | Describe the dance movements that correspond to the given music <Music_Placeholder>. Describe the dance actions that match the provided music <Music_Placeholder>. Detail the dance steps associated with the specified music <Music_Placeholder>. | <Caption_Placeholder> |

the comparison of text-to-motion on Motion-X dataset. Tab. 10 shows the comparison of motion-to-text on Motion-X dataset. Tab. 11 shows the comparison of music-to-dance on AIST++ and FineDance datasets. Tab. 12 shows the comparison of dance-to-music on AIST++ and FineDance datasets. Tab. 13 and Tab. 14 shows the comparison of motion prediction and motion in-between on Motion-X dataset. Tab. 15 shows the comparison of music-to-text, text-to-dance and text-to-music on AIST++ dataset. Tab. 16 and Tab. 17 show the comparison of motion-related tasks among different size of T5. As shown in these tables, M³GPT can achieve competitive performance with SOTAs across all evaluated tasks.

# D  Qualitative Results and Comparison with SOTA Methods

Fig. 6 presents visualizations for a variety of tasks, including text-to-motion, motion-to-text, motion prediction, motion in-between, music-to-dance, long-term dance generation, and music-text conditioned dance generation. The visualization results show that our method can generate realistic results across various motion-relevant tasks. Fig. 7 presents the qualitative results between different methods for text-to-motion and music-to-dance.

# E  Additional Experiments

**The performance of text-to-motion based on different $\lambda$.** In Tab. 18, the performance of M³GPT on Motion-X dataset is analyzed across different values of $\lambda$ for the text-to-motion task. The results indicate that as $\lambda$ increases, the model's recall precision (Top1, Top2, Top3) initially improves but subsequently declines. The optimal performance is achieved at $\lambda = 0.2$, as evidenced by the highest scores in RPrecision and modality metrics.

Further increases in $\lambda$ to 0.3 and 0.4 lead to deteriorating performance, particularly in FID and RPrecision, suggesting that excessive $\lambda$ values may result in over-regularization or reduced adaptability of the model.

Table 9: Comparison of Text-to-Motion on Motion-X dataset [29]. The arrows (↑) indicate that higher values are better. The arrows (↓) indicate that smaller values are better. **Bold** and underline indicate the best and the second best result.

| Methods | RPrecision ↑ | | | FID ↓ | MM Dist ↓ | Div ↑ | MModality ↑ |
|---|---|---|---|---|---|---|---|
| | Top1 | Top2 | Top3 | | | | |
| Real | $0.675^{\pm0.003}$ | $0.821^{\pm0.003}$ | $0.878^{\pm0.002}$ | $0.009^{\pm0.000}$ | $2.938^{\pm0.007}$ | $2.316^{\pm0.011}$ | - |
| MDM [44] | $0.472^{\pm0.008}$ | $0.616^{\pm0.005}$ | $0.704^{\pm0.003}$ | $0.161^{\pm0.006}$ | $5.404^{\pm0.031}$ | $2.234^{\pm0.015}$ | $2.241^{\pm0.043}$ |
| MLD [5] | $0.612^{\pm0.003}$ | $0.743^{\pm0.002}$ | $0.808^{\pm0.004}$ | $0.122^{\pm0.008}$ | $3.117^{\pm0.035}$ | $2.267^{\pm0.018}$ | $2.210^{\pm0.055}$ |
| T2M-GPT [55] | $0.647^{\pm0.002}$ | $0.785^{\pm0.004}$ | $0.845^{\pm0.003}$ | $0.101^{\pm0.005}$ | $3.046^{\pm0.028}$ | $2.270^{\pm0.033}$ | **$2.226^{\pm0.036}$** |
| MotionDiffuse [55] | $0.659^{\pm0.002}$ | $0.802^{\pm0.004}$ | **$0.865^{\pm0.002}$** | **$0.075^{\pm0.004}$** | $2.944^{\pm0.004}$ | $2.220^{\pm0.022}$ | $2.102^{\pm0.036}$ |
| Trained single task | $0.656^{\pm0.002}$ | $0.795^{\pm0.001}$ | $0.843^{\pm0.001}$ | $0.078^{\pm0.000}$ | $2.942^{\pm0.001}$ | $2.133^{\pm0.012}$ | $2.046^{\pm0.052}$ |
| M$^3$GPT (Pre-trained) | $0.601^{\pm0.002}$ | $0.751^{\pm0.003}$ | $0.803^{\pm0.002}$ | $0.092^{\pm0.002}$ | $2.945^{\pm0.001}$ | $2.251^{\pm0.012}$ | $2.188^{\pm0.074}$ |
| M$^3$GPT (Fine-tuned) | $0.615^{\pm0.003}$ | $0.757^{\pm0.004}$ | $0.815^{\pm0.003}$ | $0.093^{\pm0.002}$ | $2.944^{\pm0.002}$ | $2.253^{\pm0.023}$ | $2.204^{\pm0.058}$ |
| M$^3$GPT(Fine-tuned only T2M) | **$0.661^{\pm0.003}$** | **$0.804^{\pm0.004}$** | $0.861^{\pm0.003}$ | $0.076^{\pm0.002}$ | **$2.940^{\pm0.002}$** | **$2.273^{\pm0.026}$** | $2.131^{\pm0.032}$ |

Table 10: Comparison of Motion-to-Text on Motion-X [29].

| Methods | RPrecision ↑ | | | MM Dist ↓ | Bleu@1 ↑ | Bleu@4 ↑ | Rouge ↑ | CIDEr ↑ | BertScore ↑ |
|---|---|---|---|---|---|---|---|---|---|
| | Top1 | Top2 | Top3 | | | | | | |
| Real | 0.681 | 0.824 | 0.881 | 2.897 | - | - | - | - | - |
| TM2T [15] | 0.574 | 0.726 | 0.806 | 2.988 | 30.54 | **12.13** | 32.52 | 20.16 | 25.37 |
| Trained single task | 0.565 | 0.706 | 0.767 | 3.011 | 31.07 | 10.14 | 31.65 | 22.92 | 28.19 |
| M$^3$GPT (Pre-trained) | 0.627 | 0.773 | 0.834 | **2.946** | 33.31 | 11.00 | 34.10 | 24.12 | 30.96 |
| M$^3$GPT (Fine-tuned) | **0.631** | **0.783** | **0.845** | 2.950 | **34.27** | 11.50 | **34.55** | **42.93** | **31.49** |

Table 11: Comparison of Music-to-Dance on AIST++ [24] and FineDance [25].

| Methods | Music-to-Dance on AIST++ | | | Music-to-Dance on FineDance | | |
|---|---|---|---|---|---|---|
| | FID$_k$ ↓ | Div$_k$ ↑ | BAS↑ | FID$_k$ ↓ | Div$_k$ ↑ | BAS↑ |
| Real | 17.10 | 8.19 | 0.2374 | - | 9.73 | 0.2120 |
| FACT [24] | 35.35 | 5.94 | 0.2209 | 113.38 | 3.36 | 0.1831 |
| Bailando [42] | 28.16 | **7.83** | 0.2332 | 82.81 | 7.74 | 0.2029 |
| EDGE [48] | 42.16 | 3.96 | 0.2334 | 94.34 | **8.13** | 0.2116 |
| Lodge [26] | 37.09 | 5.58 | **0.2423** | **45.56** | 6.75 | **0.2397** |
| Trained single task | 75.47 | 5.57 | 0.1884 | 128.37 | 6.48 | 0.2036 |
| M$^3$GPT (Pre-trained) | 27.65 | 7.52 | 0.2105 | 92.35 | 7.67 | 0.2134 |
| M$^3$GPT (Fine-tuned) | **24.34** | 7.50 | 0.2056 | 86.47 | 7.75 | 0.2158 |

Table 12: Comparison of Dance-to-Music on AIST++ [24] and FineDance [25].

| Methods | Dance-to-Music on AIST++ | | | | | Dance-to-Music on FineDance | | | | |
|---|---|---|---|---|---|---|---|---|---|---|
| | BCS ↑ | CSD ↓ | BHS ↑ | HSD ↓ | F1 ↑ | BCS ↑ | CSD ↓ | BHS ↑ | HSD ↓ | F1 ↑ |
| Foley [11] | 96.4 | 6.9 | 41.0 | 15.0 | 57.5 | - | - | - | - | - |
| CMT [10] | 97.1 | 6.4 | 46.2 | 18.6 | 62.6 | - | - | - | - | - |
| D2MGAN [64] | 95.6 | 9.4 | 88.7 | 19.0 | 93.1 | - | - | - | - | - |
| CDCD [65] | 96.5 | 9.1 | 89.3 | 18.1 | 92.7 | - | - | - | - | - |
| LORIS [52] | **98.6** | **6.1** | 90.8 | 13.9 | 94.5 | - | - | - | - | - |
| Trained single task | 93.9 | 9.2 | 93.6 | 12.8 | 92.8 | **84.84** | 21.61 | 51.35 | 27.13 | 63.97 |
| M$^3$GPT (Pre-trained) | 93.4 | 10.9 | 93.8 | 11.5 | 94.2 | 83.16 | 19.95 | 73.65 | 23.90 | 78.12 |
| M$^3$GPT (Fine-tuned) | 93.6 | 10.1 | **94.0** | **10.6** | **94.9** | 84.10 | **18.36** | 74.66 | 23.45 | 78.23 |

Table 13: Comparison of Motion Prediction and Motion In-between on Motion-X [29].

| Methods | Motion Prediction | | | | Motion In-between | | |
| --- | --- | --- | --- | --- | --- | --- | --- |
| | FID ↓ | Diversity ↑ | ADE ↓ | FDE ↓ | FID ↓ | Diversity ↑ | ADE ↓ |
| Ground Truth | 0.009 | 2.316 | - | - | 0.009 | 2.316 | - |
| MDM [44] | 1.028 | 1.746 | 8.057 | 11.266 | 0.831 | 1.768 | 6.542 |
| Trained single task | 0.774 | 1.778 | 7.840 | 9.575 | 0.692 | 1.810 | 6.690 |
| M$^3$GPT-pretrain | 0.707 | **1.874** | 6.954 | 8.684 | **0.604** | 1.897 | 5.692 |
| M$^3$GPT-finetune | **0.682** | 1.838 | **6.898** | **8.091** | 0.612 | **1.900** | **5.649** |

Table 14: Comparison of MPJPE on Motion Prediction and Motion In-between on Motion-X [29].

| Methods on Motion-X | Motion Prediction (MPJPE ↓) | Motion In-between (MPJPE ↓) |
| --- | --- | --- |
| T2M-GPT [55] | 80.2 | 63.7 |
| MoMask [16] | 67.9 | 55.2 |
| MotionGPT [21] | 71.3 | 59.9 |
| M$^3$GPT (Instruction-tuned) | **54.2** | **51.0** |

Table 15: Comparison of Music-to-Text (A2T), Text-to-Dance (T2D) and Text-to-Music (T2A) on AIST++.

| Methods on AIST++ | Music-to-Text | | Text-to-Dance | | Text-to-Music | |
| --- | --- | --- | --- | --- | --- | --- |
| | Bleu@4 ↑ | CIDEr ↑ | R-TOP1 ↑ | FID ↓ | BCS ↑ | BHS ↑ |
| M$^3$GPT (Single task training for A2T) | 9.24 | 24.55 | - | - | - | - |
| M$^3$GPT (Single task training for T2D) | - | - | 0.541 | 0.095 | - | - |
| MusicLDM [2] | - | - | - | - | **74.5** | 73.8 |
| Mubert | - | - | - | - | 73.3 | 73.0 |
| M$^3$GPT (Instruction-tuned) | **11.95** | **28.88** | **0.588** | **0.077** | **74.5** | **74.7** |

Table 16: Comparison of Text-to-Motion and Motion-to-Text with different size of T5.

| Methods on Motion-X | LLM | Training time | Text-to-Motion | | | Motion-to-Text | | |
| --- | --- | --- | --- | --- | --- | --- | --- | --- |
| | | | R-TOP1 ↑ | FID ↓ | Div ↑ | R-TOP3 ↑ | Bleu4 ↑ | CIDEr ↑ |
| M$^3$GPT | T5-small (60M) | 5 days | 0.598 | 0.096 | 2.202 | 0.822 | 10.43 | 38.22 |
| M$^3$GPT | T5-base (220M) | 7 days | 0.615 | 0.093 | 2.253 | 0.845 | 11.50 | 42.93 |
| M$^3$GPT | T5-large (770M) | 8 days | **0.619** | **0.090** | **2.256** | **0.848** | **11.64** | **43.05** |

Table 17: Comparison of Music-to-Dance and Dance-to-Music with different size of T5.

| Methods on AIST++ | LLM | Training time | Music-to-Dance | | | Dance-to-Music | |
| --- | --- | --- | --- | --- | --- | --- | --- |
| | | | FID$_k$ ↓ | DIV$_k$ ↑ | BAS ↑ | BCS ↑ | BHS ↑ |
| M$^3$GPT | T5-small (60M) | 5 days | 28.05 | 5.96 | 0.2091 | 89.1 | 91.2 |
| M$^3$GPT | T5-base (220M) | 7 days | 24.34 | 7.50 | 0.2056 | 93.6 | 94.0 |
| M$^3$GPT | T5-large (770M) | 8 days | **23.26** | **7.54** | **0.2061** | **93.8** | **94.1** |

Table 18: Hyper-parameter analysis of $\lambda$. Comparison of Text-to-Motion on Motion-X [29] with different values of $\lambda$. For this ablation study, M$^3$GPT is trained solely on the text-to-motion task to examine the impact of $\lambda$. This study is conducted during the pre-training stage.

| Methods | RPrecision ↑ | | | FID ↓ | MM Dist ↓ | Diversity → | MModality ↑ |
| --- | --- | --- | --- | --- | --- | --- | --- |
| | Top1 | Top2 | Top3 | | | | |
| Real | $0.675^{\pm0.003}$ | $0.821^{\pm0.003}$ | $0.878^{\pm0.002}$ | $0.009^{\pm0.000}$ | $2.938^{\pm0.007}$ | $2.316^{\pm0.011}$ | - |
| $\lambda$=0.0 | $0.645^{\pm0.002}$ | $0.778^{\pm0.003}$ | $0.826^{\pm0.002}$ | $0.081^{\pm0.002}$ | $2.944^{\pm0.002}$ | $2.124^{\pm0.011}$ | $2.025^{\pm0.021}$ |
| $\lambda$=0.1 | $0.649^{\pm0.002}$ | $0.787^{\pm0.002}$ | $0.838^{\pm0.002}$ | $\mathbf{0.078}^{\pm0.002}$ | $2.944^{\pm0.003}$ | $2.128^{\pm0.022}$ | $2.039^{\pm0.049}$ |
| $\lambda$=0.2 | $\mathbf{0.656}^{\pm0.002}$ | $\mathbf{0.795}^{\pm0.001}$ | $\mathbf{0.843}^{\pm0.001}$ | $\mathbf{0.078}^{\pm0.000}$ | $2.942^{\pm0.001}$ | $\mathbf{2.133}^{\pm0.012}$ | $\mathbf{2.046}^{\pm0.052}$ |
| $\lambda$=0.3 | $0.629^{\pm0.002}$ | $0.716^{\pm0.002}$ | $0.784^{\pm0.001}$ | $0.095^{\pm0.002}$ | $\mathbf{2.941}^{\pm0.001}$ | $2.113^{\pm0.037}$ | $2.036^{\pm0.041}$ |
| $\lambda$=0.4 | $0.573^{\pm0.001}$ | $0.725^{\pm0.002}$ | $0.793^{\pm0.002}$ | $0.102^{\pm0.002}$ | $2.942^{\pm0.002}$ | $2.090^{\pm0.007}$ | $2.030^{\pm0.086}$ |

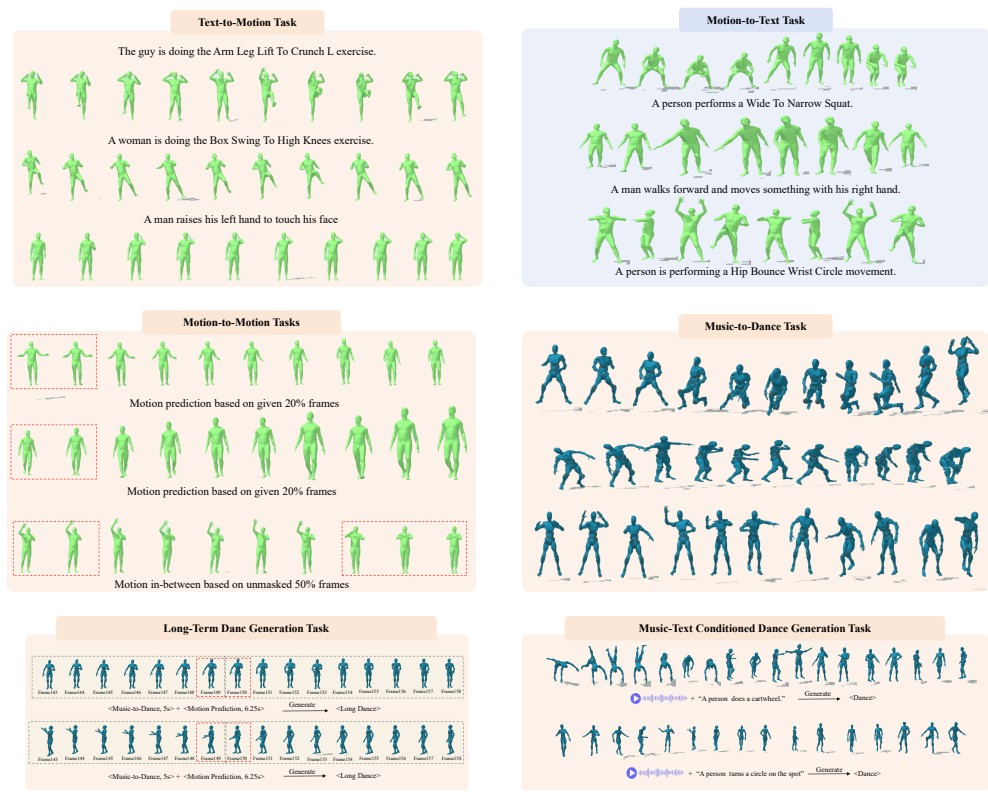

Figure 6: The qualitative results for different motion comprehension and generation tasks.

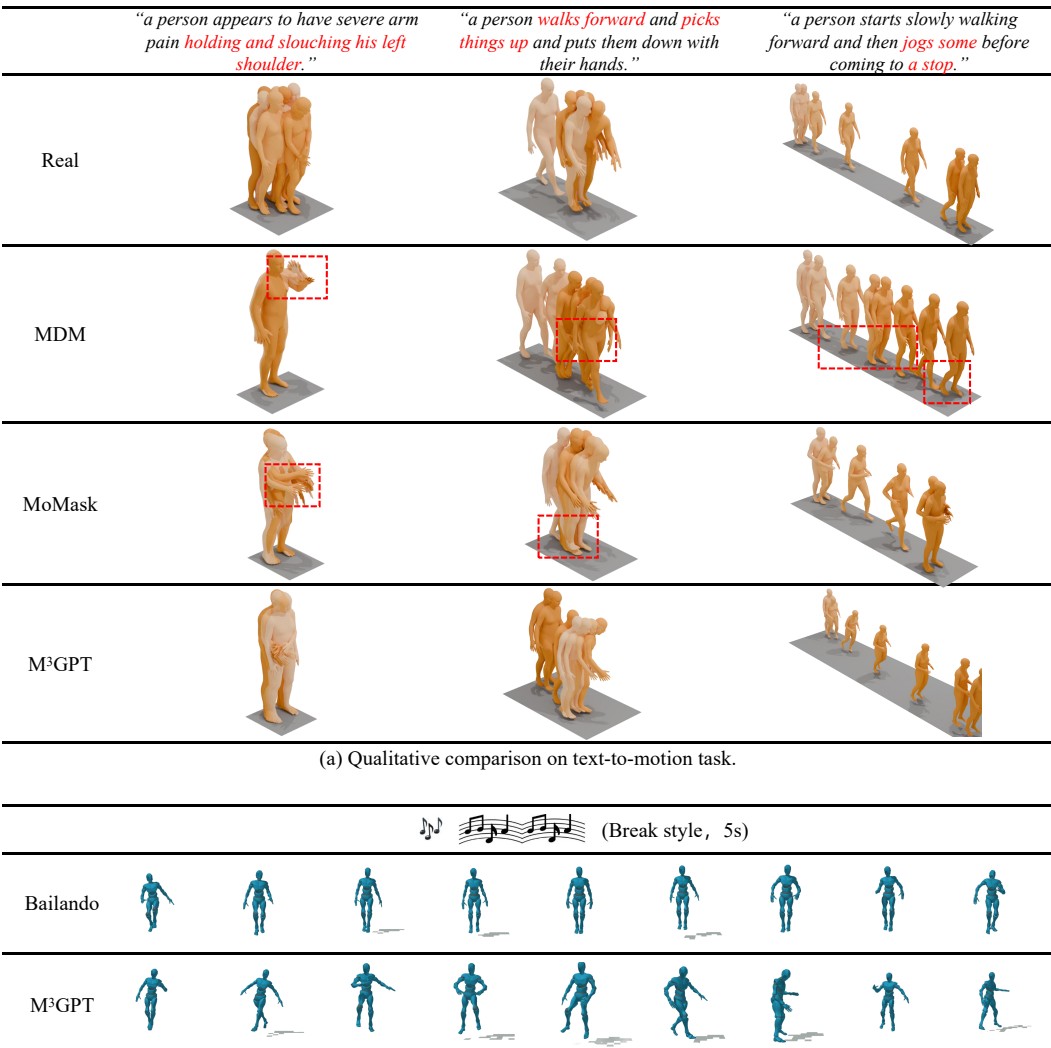

(a) Qualitative comparison on text-to-motion task.

(b) Qualitative comparison on music-to-dance task.

Figure 7: Qualitative comparisons for text-to-motion task and music-to-dance task. (a) refers to the qualitative comparison between Real, MDM, MoMask and M3GPT on text-to-motion task. The red words and boxes highlight the misaligned motions. The results demonstrate that our M3GPT shows good text understanding for motion generation. (b) refers to the qualitative comparison between Bailando and M3GPT on music-to-dance task. The input is an 5-second-long piece of music in the Break style.

