# OpenReview forum: "M$^3$GPT: An Advanced Multimodal, Multitask Framework  for Motion Comprehension and Generation"
_NeurIPS.cc/2024/Conference — NeurIPS 2024 poster_

### Official Review · Reviewer_2Gpw · 2024-07-11

**Soundness:** 3
**Presentation:** 2
**Contribution:** 2
**Rating:** 5
**Confidence:** 4

**Summary:**

This paper introduces M3GPT, an advanced Multimodal, Multitask framework for motion comprehension and generation. M3GPT operates based on three fundamental principles. Firstly, it aims to create a unified representation space for various motion-relevant modalities. We employ discrete vector quantization for multimodal control and generation signals, such as text, music, and motion/dance, enabling seamless integration into a large language model (LLM) with a single vocabulary. Secondly, it involves modeling directly in the raw motion space. This strategy circumvents the information loss associated with discrete tokenizers, resulting in more detailed and comprehensive model generation. Thirdly, M3GPT learns to model the connections and synergies among various motion-relevant tasks. Text, being the most familiar and well-understood modality for LLMs, is utilized as a bridge to establish connections between different motion tasks, facilitating mutual reinforcement.

**Strengths:**

1.     M3GPT is equipped with multimodal tokenizers capable of compressing raw multimodal data, including motion, music, and text, into a sequence of discrete semantic tokens.
2.	The paper jointly trains LLM and motion de-tokenizer, optimizing LLM in both discrete semantic space and raw continuous motion space.
3.	This Paper is well writing and the video results in Supp. are well presented.

**Weaknesses:**

1.	In Table 3, some comparation results are missed. For text to motion generation task, you could compare your method with 1. MoMask: Generative Masked Modeling of 3D Human Motions; 2. Plan, posture, and go: Towards Open-World Text-to-Motion Generation; and 3. Real-time Controllable Motion Generation via Latent Consistency Model. If it’s not suitable for the comparations, you could explain the reason.
2.	The parameterizations (dimensions) of the variable in Section 3 are all missing.
3.	In line 167, T5 is utilized as the language model backbone. While most of the previous model utilize CLIP or BERT, you should include ablation study about the text encoder.

**Questions:**

1.	Is it just a matter of migrating multimodal model to the motion domain?
2.	What new issues have been encountered in the process？
3.	What is your innovation in solving these problems?

**Limitations:**

The author has addressed the limitations. This paper focuses on modeling human body movements, excluding hands and faces details.

---

> ### Author Rebuttal · Authors · 2024-08-07
>
> **W1: In Table3, some comparation results are missed for text-to-motion task.**
>
> Thanks for your suggestions. We will add the related works mentioned, MoMask [1], PRO-Motion [2] and MotionLCM [3], in revised version.
>
> (1) To ensure fair comparison, we reproduce **MoMask**  and **MotionLCM** on Motion-X dataset  using their official code. As shown in the table below, our method achieves superior performance than MotionLCM, and comparable performance with MoMask.
>
> (2) Since **PRO-Motion** is not open-sourced, we were unable to reproduce it on Motion-X dataset, so its result is not included in Table3.
>
> |Methods (Text-to-Motion) Motion-X |  R TOP1$\uparrow$ | FID$\downarrow$ |Div$\uparrow$ |
> | ---- | ----- |----|----|
> MoMask | **0.668** | **0.074**| 2.241 |
> MotionLCM | 0.658 | 0.078 | 2.206 |
> $M^3$GPT (Instruction-tuned) | 0.615 | 0.093 | 2.253 |
> $M^3$GPT (Instruction-tuned only T2M) | 0.661 | 0.076 | **2.273**|
>
> **W2: The parameterizations (dimensions) of the variable in Section 3 are all missing.**
>
> Thanks for your suggestions.  We will add the dimensions of variable in section 3 as follows  in revised version.
>
> - $N_m$ (line 128), the size of motion codebook $\mathcal{B}_m$, is set to 512.
> - $d_m$ (line 131), the dimension of raw motion features, is 263.
> - $d$ (line 133), the dimension of embeddings in motion codebook, is set to 512.
> - $d_a$ (line 149), the dimension of raw music, is 128.
> - $T_m$ (line 131) and $T_a$ (line 149) denotes the sequence length of motion and music, respectively. So their values depend on the input sequence.
> - $L_m$ (line 133), the sequence length of diverse motion tokens, is $T_m/4$. Here, 4 is the downsampling rate of motion tokenizer.
> - $L_a$ (line 150), the sequence length of diverse music tokens, is $T_a/128$. Here, 128 is the downsampling rate of music tokenizer.
>
> **W3: T5 is utilized as the language model backbone. While most previous model utilize CLIP or BERT, you should include ablation study about the text encoder.**
>
> Following your suggestions, We add ablation study on CLIP and BERT as text encoder. Specifically, we use T5, CLIP and BERT as text encoders respectively, each combined with T5 decoder. We then conducted experiments on the text-to-motion task using these configurations. As shown in below table, T5 encoder can achieve the best performance, validating its superiority over CLIP and BERT in our framework.
>
> Methods (Text-to-Motion) Motion-X  |R TOP1$\uparrow$   |FID$\downarrow$ |Div$\uparrow$|
> | ---- | ---- | ---- | ---- |
> CLIP | 0.641 | 0.088 | 2.110
> BERT | 0.652 | 0.083 | 2.124
> T5   | **0.656** | **0.078** | **2.133**
>
> **Q1: Is it just a matter of migrating multimodal model to the motion domain?**
>
>  $M^3$GPT is not simply migrating multimodal model to motion domain. Unlike most multimodal models such as BLIP2, LLaVA, Ferret, which primarily focus on understanding tasks, $M^3$GPT takes a step further by incorporating multimodal tokenizers and de-tokenizers, achieving both motion **comprehension and generation** across **text, motion/dance and music** modalities. Additionally, $M^3$GPT **jointly optimizes LLM and motion de-tokenizer**, further enhancing its capabilities in motion generation.
>
> **Q2: What new issues have been encountered in the process?**
>
> We have encountered two new issues:
> 1. Within the multi-task framework, music-to-dance task poses greater challenges than text-to-motion. This  difficulty arises because both input and output modalities of music-to-dance are unfamiliar to LLMs, and there is limited training data available for this task. Therefore, **effectively leveraging text-to-motion task to enhance more complex music-to-dance task is a critical issue that needs to be addressed.**
> 2. The multi-stage training leads to error accumulation in motion generation, due to the combined effects of LLM's prediction and de-tokenizer. Therefore, **mitigating  error accumulation from multi-stage training is another critical issue that needs to be addressed.**
>
> **Q3: What is your innovation in solving these problems?**
>
> 1. To solving the first issue outlined in Q2, we design two strategies.  (1) a **shared tokenizer** for motion and dance data, and (2) two **auxiliary tasks: music-to text and text-to-dance**. Through these auxiliary tasks, $M^3$GPT implicitly learns to decompose complex music-to-dance into simpler music-to-text and text-to-dance tasks. Also, the shared tokenizer ensures that text-to-dance and text-to-motion tasks occupy the same matching space, enabling mutual reinforcement. In this way, $M^3$GPT builds the connection between text-to-motion and music-to-dance, leveraging text-to-motion to enhance  its music-to-dance capabilities.
> 2. To solving the second issue outlined in Q2, we **jointly optimize LLM and motion de-tokenizer** during training. This strategy can help mitigate error accumulation caused by multi-stage training, thereby enhancing the model's motion generation capabilities.

---

> ### Author Response · Authors · 2024-08-12
> **Any questions about the rebuttal**
>
> Dear Reviewer 2Gpw:
>
> As the rebuttal period is ending soon, please let us know whether your concerns have been addressed or not, and if there are any further questions.

---

### Official Review · Reviewer_KMVC · 2024-07-11

**Soundness:** 3
**Presentation:** 3
**Contribution:** 3
**Rating:** 6
**Confidence:** 4

**Summary:**

The paper presents M3GPT, which creates a unified representation space for different motion-relevant modalities, including text, music, and motion/dance. The framework employs discrete vector quantization for these modalities, enabling integration into a large language model (LLM) with a single vocabulary. The model operates directly in the raw motion space to avoid information loss associated with discrete tokenizers. Extensive experiments highlight M3GPT’s superior performance across various motion tasks and its zero-shot generalization capabilities.

**Strengths:**

- The paper introduces a novel approach to integrate multiple modalities (text, music, motion) within a single framework, addressing a significant gap in existing research. The unified representation space and the strategy of using text as a bridge to connect different modalities are theoretically sound and contribute to the field of multimodal learning.

- M3GPT handles six core tasks related to motion comprehension and generation, demonstrating versatility and robustness. The experiments show that M3GPT achieves competitive performance across multiple tasks.

- The model's ability to perform well in zero-shot scenarios, such as long-term dance generation and music-text conditioned dance generation, is particularly impressive and valuable for practical applications.

**Weaknesses:**

- The proposed framework is complex and may require significant computational resources to implement and extend.

- While the experimental results are promising, the paper could benefit from a more detailed discussion of the evaluation metrics and benchmarks used to assess performance. A more extensive comparison with a broader range of baselines could strengthen the argument for its superiority. Also, the paper would benefit from a more in-depth discussion on the potential real-world applications and limitations of M3GPT.

**Questions:**

- Could you provide more details on the training process, including the computational resources required and the time taken to train M3GPT?

- Have you conducted any ablation studies to understand the contribution of each component of M3GPT to the overall performance? If so, could you include those results?

- Can you provide more examples or case studies where M3GPT has been applied to real-world scenarios? What challenges were encountered?

**Limitations:**

Limitations have been discussed.

---

> ### Author Rebuttal · Authors · 2024-08-07
>
> **W1: The proposed framework is complex and may require significant computational resources to implement.**
>
> To ensure reproducibility, we will open-source the training and inference code of proposed $M^3$GPT framework.
>
> For computational resources, please  refer to **reply to Q2 in Global Response**.
>
>
> **W2-1: The paper could benefit from a more detailed discussion of evaluation metrics and benchmarks.**
>
> Thanks for your suggestions. We will add these discussion in final version.
>
> **(1) Evaluation Metrics.** The table below shows the evaluation metrics for each task.
>
> |Task| Text-to-Motion (T2M) |Motion-to-Text (M2T)| Motion Prediction/In-Between| Music-to-Dance (A2D) | Dance-to-Music (D2A)|
> | ----| ----- | ----- | ----- |----- |----|
> |Metrics| FID, R-Precision, Div| R-Precision, Bleu, CIDEr, Rouge| FID, Div, MPJPE| FID$_k$, Div$_k$, BAS|  BCS, BHS|
>
> - FID: Measures distance between the distributions of generated and real motion.
> - R-Precision: Measures retrieval accuracy of generated motion/text relative to input, reflecting semantic similarity.
> - Div: Measures diversity of generated motions.
> - Bleu, Rouge, CIDEr: Measures accuracy, matching degree and semantic consistency between generated and real text.
> - FID$_k$, Div$_k$: FID and Div calculated using kinetic features.
> - MPJPE: Measures the mean per joint position error between generated and real motion.
> - BAS: Computes beat alignment degree between generated dance and input music.
> - BCS/BHS: Measures convergence/alignment between generated and real music.
>
> **(2) More Benchmarks.** The tables below adds two benchmarks. As shown in below table, and Table3/4 of main paper, we observe that:
> - Text-to-Motion: $M^3$GPT achieves comparable FID and best R-Top1 and Div score, showing high-quality motion generation with strong semantic alignment and diversity.
> - Motion-to-Text: $M^3$GPT achieves  best R-precision and CIDEr score, indicating fluent and semantically consistent motion descriptions generation.
> - Motion Prediction and Inbetween: $M^3$GPT achieves  best FID, Div and MPJPE score, reflecting superior motion prediction quality, diversity and precision.
> - Music-to-Dance (Table 4):  $M^3$GPT achieves comparable Div/BAS and best FID$_k$ score, indicating high-quality dance generation with good beat.
> - Dance-to-Music (Table 4), $M^3$GPT achieves  comparable BCS and best BHS score, indicating improved semantic alignment of music with input dance.
>
>
> |Methods on Motion-X| T2M (R TOP1$\uparrow$   \| FID$\downarrow$ \| Div$\uparrow$) | M2T (R TOP3$\uparrow$ \| Bleu@4$\uparrow$ \|CIDEr$\uparrow$) | Motion Prediction (FID$\downarrow$ \| Div$\uparrow$\| MPJPE$\downarrow$)| Motion In-between (FID \| Div \| MPJPE)
> | ---- | ----- | ----- | ----- |----- |
> |MotionLCM| 0.658 \| 0.078 \| 2.206| -| -| -|
> |MotionGPT| 0.659 \| 0.078 \| 2.166 | 0.840 \| 11.21 \| 31.36 | 0.701 \| 1.818 \| 71.3 | 0.648 \| 1.875 \| 59.9
> |$M^3$GPT| 0.661 \| 0.076 \| 2.273 | 0.845 \| 11.50 \| 42.93 |0.682 \| 1.838 \| 54.2 | 0.612 \| 1.900 \| 51.0
>
>
>
> **W2-2: The paper could benefit from a more in-depth discussion on potential real-world applications and limitations of $M^3$GPT.**
>
> **(1) Potential applications:** $M^3$GPT has may real-world applications, such as **animation creation, visual digital human control, music-based  dance choreography creation**. Taking animation creation as a example, traditional manual methods adjusts   character joints frame-by-frame, which is extremely time-consuming and labor-intensive. With $M^3$GPT, one can simply provide a textural description of desired motion, and $M^3$GPT will automatically generate the corresponding animation, largely reducing labor and time cost.
>
> **(2) Limitations:** $M^3$GPT focus on modeling human body movements, but currently lacks the capability to handle hands. As a result, $M^3$GPT cannot address speech-driven motion generation, which involves gestural rather than body movement.
>
> **Q1: Could you provide more details on the training process, including the computational resources required and the time taken to train $M^3$GPT?**
>
> Yes, please refer to our **reply to Q2 in Global Response**.
>
> **Q2: Have you conducted any ablation studies to understand the contribution of each component of $M^3$GPT to the overall performance**
>
> Yes, in Table 2 of main paper, we have conducted ablation studies to show the effectiveness of each component of $M^3$GPT. Specifically, $M^3$GPT includes 3 major components designed to improve performance: **(1)** Introduction of two auxiliary tasks, music-to-dance  (A2D) and text-to-music (T2D). **(2)** Joint optimization of LLM and motion de-tokenizer. **(3)** Instruction-tuning. As shown in the table below, each component contributes to the overall performance of $M^3$GPT, specifically for the complex music-to-dance task.
>
>
> |Methods|  T2M on Motion-X (R TOP1   \| FID    \| Div) | A2D on AIST++ (FID_k \| Div_k   \| BAS)
> | ---- | ----- | ----- |
> |$M^3$GPT (Pretrained without A2T and T2D)| 0.547 \| 0.104 \| 2.099 | 37.14 \| 7.61 \| 0.200
> |$M^3$GPT (Pretrained without joint optimization)| 0.598 \| 0.089 \| 2.218 | 32.71 \| 7.43 \| 0.209
> |$M^3$GPT (Pretrained)| 0.601 \| 0.092 \| 2.251 | 27.65 \| 7.52 \| 0.210 |
> |$M^3$GPT (Instruction-tuned)| 0.615 \| 0.093 \| 2.253 | 24.34 \| 7.50 \| 0.205
> |$M^3$GPT (Instruction-tuned only single task )| **0.661** \| **0.076** \| **2.273** | **23.01** \| **7.85** \| **0.226**
>
> **Q3: Can you provide more examples or case studies where $M^3$ has been applied to real-world scenarios? What challenges were encountered?**
>
> We have applied $M^3$GPT to several real-world scenarios, including animation creation, dance choreography creation, music creation from dance.
>
> The main challenge is generating long sequence (longer than 5 minutes), where the quality of  generated content degrades over time. We leave the generation of high-quality  long-sequence (longer than 5 minutes) as a future research  direction.

---

> ### Author Response · Authors · 2024-08-12
> **Any questions about the rebuttal**
>
> Dear Reviewer KMVC:
>
> As the rebuttal period is ending soon, please let us know whether your concerns have been addressed or not, and if there are any further questions.

---

### Official Review · Reviewer_Wkeh · 2024-07-11

**Soundness:** 3
**Presentation:** 3
**Contribution:** 3
**Rating:** 5
**Confidence:** 5

**Summary:**

This paper presents an advanced Multimodal, Multitask framework for Motion comprehension and generation. It aims to create a unified representation space for various motion-relevant modalities and model the connections and synergies among different motion tasks. M3GPT consists of multimodal tokenizers and a motion-aware language model, and is trained through a multimodal pre-training + instruction-tuning pipeline. The model is evaluated on multiple motion-relevant tasks and datasets, and shows competitive performance and strong zero-shot generalization capabilities.

**Strengths:**

(1) Unifies multiple motion-related tasks in a single model, achieving bidirectional alignment.

(2) Achieves competitive performance across multiple tasks and datasets and strong zero-shot generalization capabilities.

(3) Explores motion related music2text and text2dance as auxiliary tasks.

**Weaknesses:**

(1) For supplementary videos, it's hard to judge the quality of music-motion bidirectional generations without combining them in one video; also, long-term dance generation on aist++ dataset is only a little longer than 5s.

(2) The method of Joint optimization of LLM and motion de-tokenizer lacks important details. In line 194-195, I wonder how the gradient from de-tokenizer backprogate to LLM if author use softmax to choose top-1 token.

(3) For motion prediction and In-between tasks, feature-level metrics like FID are not enough, pose-level metrics like MPJPE are needed for comparison.

**Questions:**

(1) Did authors try different LLM architecture or different size of T5, what is the conclusion and the training time for T5 base?

(2) Did you evalute two auxiliary tasks music2text or text2dance quantitatively or qualitatively?

(3) For zero-shot tasks, can they combined together for more application? For instance, generating a long dance while we specify text prompt for one segment in the middle of sequence. Also, how about zero-shot text2music?

(4) For tab.4, what is the result of Instruction-tuned only on AIST++ or FineDance of which the setting is similar to the last row in tab.3.

(5) How did you implement motion in-between in pretrained stage use the equation(4)? More specifically, whether your model is conditioned on future target frame, if so, the equation is not precise enough.

(6) Is there any special token for motion and music in the unified token vocabulary, like start/end of motion/music token? If not, how to ensure the pretrained model generate desired modality? For example, I wanna directly generate dance conditioned on music, but actually the model generate text token.

(7) The statistics of music2text and text2dance paired dataset are missing.


I would like to raise the score if authors can well address my concerns questions.

**Limitations:**

N.A

---

> ### Author Rebuttal · Authors · 2024-08-07
>
> **W1: For supplementary videos, it's hard to judge the quality of music-motion generations without combining them in one video; Long-term dance generation on aist++ is only a little longer than 5s.**
>
> (1) Due to the rebuttal period restrictions on video uploads, we will present music and dance in one video for clearer evaluation in revised version.
>
> (2) Regarding long-term dance generation, AIST++ comprises musics that are slightly longer than 5s. So the generated dances on AIST++ are relatively short. To showcase our model’s capability for long-term dance generation, we have included a dance generation of over 90s on FineDance in supplementary video.
>
> **W2: How the gradient from de-tokenizer backpropagate to LLM?**
>
> In implementation, we do not directly backpropagate de-tokenizer gradient to LLM. Instead, with the goal of minimizing L1 loss between the predicted and real motion, we search for the motion's token sequence that could minimize this L1 loss in original motion space. As the motion de-tokenizer continuously optimizes, the target motion's token sequence, which supervises LLM training,  dynamically changes. This dynamic adjustment reduces L1 loss progressively, achieving joint optimization.
>
> **W3: For motion prediction and in-between tasks, pose-level metrics like MPJPE are needed for comparison.**
>
> In the tables below, we add MPJPE as a metric. Our method still outperforms existing works on MPJPE metric in both tasks.
>
>
> |Methods on Motion-X| Motion Prediction (MPJPE$\downarrow$)| Motion In-between ( MPJPE) |
> | ---- | ----- | ----- |
> |T2M-GPT | 80.2| 63.7 |
> |MoMask |67.9 | 55.2|
> |MotionGPT | 71.3| 59.9|
> | M3GPT (Instruction-tuned)| **54.2**|**51.0** |
>
>
>
> **Q1: Did authors try different LLM architecture or different size of T5, what is conclusion and training time for T5?**
>
>  Yes, Please refer to our **reply to Q1 in Global Response**.
>
> **Q2:  Did you evalute two auxiliary tasks music2text (A2T) or text2dance (T2D) quantitatively or qualitatively?**
>
> (1) For quantitative evaluation, we report the *Bleu@4* and *CIDEr* for A2T, and *R Top1* and *FID* for T2D. As shown in the tables below, our method consistently outperforms single task training.
>
> (2) For qualitative evaluation, **Figure 4 of uploaded PDF** visualizes the generation of text2dance. Compared to the singe-task model, $M^3$GPT generates more realistic dance aligned with input music.
>
> |Methods (Music-to-Text) AIST++|Bleu@4$\uparrow$ | CIDEr$\uparrow$|
> |----|----|----|
> |A2T (Single task)| 9.24 | 24.55 |
> |$M^3$GPT (Instruction-tuned)| **11.95**| **28.88**
>
> |Methods (Text-to-Dance) AIST++|R Top1$\uparrow$| FID$\downarrow$|
> |----|----|----|
> |T2D (Single task)| 0.541 | 0.095
> |$M^3$GPT (Instruction-tuned)| **0.588** | **0.077**
>
> **Q3: For zero-shot tasks, can they combined together for more application? For instance, generating a long dance while we specify text prompt for one segment in the middle of sequence. How about zero-shot text2music?**
>
> (1) Our model can perform additional zero-shot tasks. It can generate a long dance while specifying text prompt for one segment, as visualized in **Figure 3 of uploaded PDF**. This application integrates music-to-dance, zero-shot **music+dance-to-dance**,  and zero-shot **music+dance+text-to-dance** tasks. As shown in that Figure 3, the generated dance is coherent and consistently aligning the input text prompt.
>
> (2) Our model can perform zero-shot text-to-music. The table below shows a quantitative evaluation on text-to-music task. Even without training on text-to-music task, $M^3$GPT can obtain comparable even superior performance over existing methods.
>
> |Methods (Text-to-Music) AIST++|BCS$\uparrow$|BHS$\uparrow$|
> |----|----|----|
> |MusicLDM| **74.5** | 73.8
> |Mubert|73.3 |73.0
> |$M^3$GPT (Instruction-tuned)| **74.5** | **74.7**
>
> **Q4: For tab.4, what is the result of Instruction-tuned only on AIST++ or FineDance?**
>
>  Please refer to our **reply to Q3 in Global Response**.
>
>
> **Q5:  How did you implement motion in-between in pretrained stage use the equation(4)?**
>
>  Motion in-between can be implemented using Equation 4. Formally, given a motion sequence $\boldsymbol{m} = \left(m_1, \dots, m_N\right)$, motion in-between task uses the first $n_1$  and last $n_2$ frames, $\boldsymbol{m_1}=\left(m_1, \dots, m_{n1}, m_{N-n2+1}, \dots, m_{N}\right)$, to predict intermediate frames $\boldsymbol{m_2}=\left(m_{n1+1}, \dots, m_{N-n2}\right)$. So the future frames in $\boldsymbol{m_1}$ are only used as **input**, and the **target** frames are the middle frames in $\boldsymbol{m_2}$.
>
> In  implementation, $\boldsymbol{m_1}$ is fed into the motion tokenizer to produce a token sequence, serving as **source input** of LLM ($\boldsymbol{q_s}$). Also, $\boldsymbol{m_2}$ is fed into the motion tokenizer to produce a token sequence, serving as **target output** of LLM ($\boldsymbol{q_t}$). Thus motion in-between can be formulated as $p_{\theta}\left(\boldsymbol{q}_t^i |\boldsymbol{q}_t^{<i}, \boldsymbol{q}_s \right)$ (Equation 4).
>
> **Q6: Is there any special token for motion and music in the unified token vocabulary, like start/end of motion/music token? If not, how to ensure the pretrained model generate desired modality?**
>
>  Yes. There are special tokens for motion and music, such as *<start_of_motion>* and *<start_of_music>*. These special start and end tokens control the beginning and end of the model's decoding process.
>
> Additionally, we use **task-specific instruction** to control model to generate desired modality. For example, the instruction *Generate a motion for the caption* is used for text-to-motion task; Instruction *Generate a music based on the dance* is used for dance-to-music task.
>
>
> **Q7: The statistics of music2text (A2T) and text2dance (T2D) paired dataset are missing.**
>
> Please refer to our **reply to Q4 in Global Response**.

---

> ### Author Response · Authors · 2024-08-12
> **Any questions about the rebuttal**
>
> Dear Reviewer Wkeh:
>
> As the rebuttal period is ending soon, please let us know whether your concerns have been addressed or not, and if there are any further questions.

---

### Official Review · Reviewer_ZwnC · 2024-07-13

**Soundness:** 3
**Presentation:** 3
**Contribution:** 2
**Rating:** 5
**Confidence:** 4

**Summary:**

In this paper, the authors introduce $M^3$GPT, a multimodal multitask framework designed for both motion comprehension and generation. Utilizing discrete vector quantization, $M^3$GPT establishes a discrete semantic representation space for various modalities. To avoid information loss during discrete de-tokenization, $M^3GPT$ jointly trains the large language model (LLM) and motion de-tokenizer, thereby optimizing the LLM within both the discrete semantic space and the continuous raw motion space. Furthermore, the authors construct paired text descriptions for music and devise two supplementary tasks music-to-text and text-to-dance, which facilitate the alignment of music and dance within the text embedding space. The efficacy of this approach is demonstrated through experiments across a range of motion-related tasks, and showcases zero-shot generalization capabilities for highly challenging tasks.

**Strengths:**

1. The motivation behind investigating the integration of text, music, and motion for motion comprehension and generation is convincing.
2. The proposed joint optimization of the large language model (LLM) and motion de-tokenizer, along with the synergy learning strategy incorporating auxiliary tasks, demonstrates potential in enhancing multimodal motion generation.

**Weaknesses:**

1. The effectiveness of combining text, music, and motion modalities in the $M^3$GPT framework is not thoroughly evaluated. Although the authors present the $M^3$GPT capabilities in handling zero-shot tasks. However, there is still doubt about how can text-to-motion or music-to-dance tasks benefit from the multimodal framework.

2. Some key experiments are missing.

   a. In Table 3, the authors evaluate proposed $M^3$GPT on text-to-motion dataset Motion-X on 4 tasks. However, MotionGPT[1] can also perform the tasks while the comparison between MotionGPT[1] is missing.

   b. Besides motion prediction lacks comparison with models like T2M-GPT[2], since the auto-regressive models are better at prediction.

   c. Motion inbetweening lacks comparison with the state-of-the-art models like MoMask[3] capable of temporal inpainting which can handle motion inbetweening tasks.

3. The qualitative comparison of generated motions by the proposed $M^3$GPT framework from text or music is absent. The inclusion of detailed qualitative results is crucial for providing a comprehensive understanding of the naturalness and overall realism of the generated gestures.


Minor:

1. In Figure 2, there is an error in drawing graphics in the Motion Codebook part.

[1] Jiang, Biao, et al. "Motiongpt: Human motion as a foreign language." Advances in Neural Information Processing Systems 36 (2024).

[2] Zhang, Jianrong, et al. "Generating human motion from textual descriptions with discrete representations." Proceedings of the IEEE/CVF conference on computer vision and pattern recognition. 2023.

[3] Guo, Chuan, et al. "Momask: Generative masked modeling of 3d human motions." Proceedings of the IEEE/CVF Conference on Computer Vision and Pattern Recognition. 2024.

**Questions:**

1. I am curious about the long-duration dance generation mentioned in Page 6, especially if the LLM can maintain coherence and consistency in generating long sequences or this long-duration generation is achieved by the consistency during de-tokenization.

2. In Table 4, the authors evaluate music-to-dance on two datasets AIST++ and FineDance, $M^3$GPT achieves state-of-the-art performance on AIST++, but the performance on FineDance is not that competitive. Could the authors provide more insights into the reasons behind this performance gap between the two datasets?

3. Have authors evaluate different sizes of the $M^3$GPT model? Given the added modalities and tasks, it would be interesting to understand how the model scales with respect to performance and computational requirements.

**Limitations:**

The authors have discussed limitations.

---

> ### Author Rebuttal · Authors · 2024-08-07
>
> **W1: There is still doubt how can text-to-motion or music-to-dance tasks benefits from multimodal framework.**
>
> We argue that text-to-motion and music-to-dance tasks benefits from multimodal framework in two main aspects:
> 1. **A shared tokenizer for motion and dance data:** The shared tokenizer inherently broadens the scale and diversity of the training data, allowing this tokenizer to learn more generalizable representations for both motion and dance data.
> 2. **Constructing two auxiliary tasks, music-to-text (A2T) and text-to-dance (T2D):** Through auxiliary tasks, $M^3$GPT implicitly learns to decompose the complex music-to-dance task into simpler A2T and T2D tasks. Also, with a shared motion tokenizer, text-to-dance and text-to-motion tasks reinforce each other within the same matching space. In this way, $M^3$GPT builds the synergies between  music-to-dance and text-to-motion, facilitating mutual reinforcement.
>
> As shown in the table below, the shared tokenizers and auxiliary tasks generally lead to gains in both tasks, validating their effectiveness.
>
> |Methods  | Text-to-Motion on Motion-X (R TOP1$\uparrow$ \| FID$\downarrow$ \| Div$\uparrow$) | Music-to-Dance on AIST++ (FID_k$\downarrow$ \| Div_k$\uparrow$ \| BAS$\uparrow$)|
> | ---- | ----- | ----- |
> |Single Task | 0.638 \| 0.095 \| 2.101 | 80.41 \| 5.51 \| 0.1882
> |Single Task with shared tokenizer| 0.656 \| 0.078 \| 2.133 | 75.47 \| 5.57 \| 0.1884
> |$M^3$GPT (without T2D and A2T)| 0.547 \| 0.104 \| 2.099 | 37.14 \| 7.61 \| 0.2005
> |$M^3$GPT (Pre-trained) | 0.601 \| 0.092 \| 2.251 | 27.65 \| 7.52 \| 0.2105
> |$M^3$GPT (Instruction-tuned only single task) | **0.661** \| **0.076** \| **2.273** |  **23.01** \| **7.85** \| **0.2261**
>
>
> **W2-a: Comparison with MotionGPT on Motion-X is missing.**
>
> The table below presents the results of MotionGPT on motion-X. Our method outperforms MotionGPT across all 4 tasks on motion-X.
>
> |Methods on Motion-X| Text-to-Motion (R TOP1$\uparrow$   \| FID$\downarrow$ \| Div$\uparrow$)| Motion-to-Text (R TOP3$\uparrow$ \| Bleu@4$\uparrow$ \| CIDEr$\uparrow$) | Motion Prediction (FID$\downarrow$ \|Div$\uparrow$ \| MPJPE $\downarrow$)| Motion In-between (FID \| Div \| MPJPE)
> | ---- | ----- | ----- | ----- |----- |
> |MotionGPT| 0.659 \| 0.078 \| 2.166 | 0.840 \| 11.21 \| 31.36 | 0.701 \| 1.818 \| 71.3 | 0.648 \| 1.875 \| 59.9
> |$M^3$GPT| **0.661** \| **0.076** \| **2.273** | **0.845** \| **11.50** \| **42.93** | **0.682** \| **1.838** \| **54.2** | **0.612** \| **1.900** \| **51.0**
>
>
> **W2-b: Lacking comparison with T2M-GPT on motion prediction task.**
>
> The table below presents the results of T2M-GPT on motion prediction task. Our method largely outperforms T2M-GPT.
>
> |Methods (Motion Prediction) Motion-X|  FID $\downarrow$   | Div $\uparrow$ | MPJPE $\downarrow$
> | ---- | ----- | ----- | ----- |
> |T2M-GPT| 0.814 | 1.755 | 80.2
> |$M^3$GPT| **0.682** | **1.838** | **54.2**
>
> **W2-c: Lacking comparison with MoMask on motion In-between task.**
>
> The table below presents the results of MoMask on motion In-between  task. Our method outperforms MoMask in motion In-between task.
>
> |Methods (Motion In-between) Motion-X|  FID $\downarrow$   | Div $\uparrow$ | MPJPE $\downarrow$
> | ---- | ----- | ----- | ----- |
> |MoMask| 0.626 | 1.884 | 55.2
> |$M^3$GPT| **0.612** | **1.900** | **51.0**
>
> **W3: The qualitative comparison of generated motions by the proposed $M^3$GPT framework from text or music is absent.**
>
> **(1) Figure 1 of uploaded PDF file** visualizes generated motions from text. We compare our generations with MDM and MoMask. As seen, our model generates motions of higher quality, with unrealistic motions from MDM and MoMask highlighted in red.
>
> **(2) Figure 2 of uploaded PDF file** visualizes generated dances from music. We compare our generations with Bailando. As seen, the dances generated by our $M^3$GPT are more danceable.
>
> **Q1: Curious about the long-duration dance generation.**
>
> We maintain coherence and consistency in generating long dance sequences during **LLM prediction phase**. For long-duration dance generation, we sequentially generate 5-second dance segments. Each dance segment is generated using corresponding music and **previously generated dance segments as control conditions**. This strategy ensures that each newly generated dance segment aligns seamlessly with the preceding ones, maintaining overall coherence.
>
> **Q2: Could the authors provide more insights into the reasons behind the performance gap between AIST++ and FineDance datasets?**
>
>  The main reason is that **the training and testing settings on FineDance are inconsistent  in our work, whereas they are consistent in other works**.
>
> Existing works typically train and test **separately** on  AIST++ and FineDance—using  5-second clips for AIST++ and 30-second clips for FineDance. In contrast, we develop a **unified multitask model across datasets**. To maintain consistency, we slice  AIST++ and FineDance data into 5-second clips for unified training, aligning with the 5 to 10-second range of motion-X data. During testing, we generate longer 30-second clips on FineDance to compare with existing methods. This inconsistency on FineDance could lead to  performance drop compared to existing models.
>
> To further validate our method, we finetune $M^3$GPT on FineDance using 30-second clips. As shown in the table below, $M^3$GPT can achieve competitive, even superior performance compared to existing methods, demonstrating its potential on FineDance.
>
> |Methods (Music-to-Dance) FineDance|  FID$_k$$\downarrow$   | Div$_k$$\uparrow$ | BAS$\uparrow$
> | ---- | ----- | ----- | ----- |
> |EDGE | 94.34 | 8.13 | 0.2116
> |Lodge | 45.56 | 6.75 | **0.2397**
> |$M^3$GPT |86.47 | 7.75 | 0.2158
> |$M^3$GPT(Instruction-tuned with 30-second clips)| **42.66** | **8.24** | 0.2231
>
>
> **Q3: Should evaluate different sizes of $M^3$GPT.**
>
> Please refer to our **reply to Q1 in Global Response**.

---

> ### Author Response · Authors · 2024-08-12
> **Any questions about the rebuttal**
>
> Dear Reviewer ZwnC:
>
> As the rebuttal period is ending soon, please let us know whether your concerns have been addressed or not, and if there are any further questions.

---

### Author Rebuttal · Authors · 2024-08-07

## **Global Response**
We sincerely thank all reviewers and ACs for reviewing our work. Some common questions are answered.

### **Q1: The evaluations for different size of T5 (Reviewers #ZwnC, #Wkeh, #KMVC)**

We conduct experiments on different sizes of T5: T5-small (60 M), T5-base (220 M), T5-large (770 M). As shown in the tables below, as the model size increasing, the performance on each task improves, but the training time  also increases. Considering the trade-off between performance and computational cost, we use T5-Base model in $M^3$GPT.

|Methods  on Motion-X | LLM  | Training time |Text-to-Motion (R TOP1$\uparrow$ \| FID$\downarrow$ \| Div$\uparrow$)| Motion-to-Text ( R TOP3$\uparrow$ \| Bleu@4$\uparrow$ \| CIDEr$\uparrow$)
| ---- | ---- | ----| ----| ----|
|  $M^3$GPT | T5-small (60M) | 5 days |0.598 \| 0.096 \| 2.202 |0.822 \| 10.43 \| 38.22
|  $M^3$GPT | T5-base (220 M) | 7 days| 0.615 \| 0.093 \| 2.253 | 0.845 \| 11.50 \| 42.93
|  $M^3$GPT | T5-large (770 M)| 8 days| **0.619** \| **0.090** \| **2.256** | **0.848** \| **11.64** \| **43.05**


|Methods on AIST++| LLM | Training time| Music-to-Dance (FID$_k$$\downarrow$   \| Div$_k$$\uparrow$ \| BAS$\uparrow$) | Dance-to-Music (BCS$\uparrow$  \| BHS$\uparrow$) |
| ---- | ----- | ----- | ----- |----|
| $M^3$GPT | T5-small (60M) | 5 days |28.05 \| 5.96 \| 0.2091 |89.1 \| 91.2
|  $M^3$GPT  | T5-base (220M)| 7 days| 23.34 \| 7.50 \| 0.2056 | 93.6 \| 94.0
| $M^3$GPT  | T5-large (770M)| 8 days| **23.26** \| **7.54** \| **0.2061** | **93.8** \| **94.1**

### **Q2: More details for training process and computational resources. (Reviewer #KMVC)**

Our training process is divided into three stages: Multimodal Tokenizers Training (Stage1); Modality-Alignment Pre-training (Stage2); Instruction Fine-tuning (Stage3). All experiments are conducted on a machine with 8 A40 (48G) GPUs. Detailed implementation can be found in Lines 276-285 of main paper. Specifically:

- Stage1: We combine Motion-X (64867 motion samples), AIST++ (952 dance samples) and FineDance  (177 dance samples) to train a motion tokenizer. This stage involves 50K training steps, with a batch size of 1000 and a learning rate of 1e-4. **Stage1 takes ~2.5 days to complete.**
- Stage2: We train on 6 main tasks and 2 auxiliary tasks. The detailed statics of training data is shown in the table below.  This stage involves 1000K training steps, with a batch size of 56 and a learning rate of 2e-4. **Stage2 takes ~5.5 days to complete.**
- Stage3: We use 200 instruction templates to construct instruction samples for finetuning the model. This stage involves 200K training steps, with a batch size of 48 and a learning rate of 1e-4. **Stage3 takes ~1.5 days to complete.**

|Tasks| T2M | M2T | Motion Prediction/In-between| A2D | D2A | A2T | T2D|
| ---- | ---- | ----| ----| ----| ----|----|----|
|Training dataset| Motion-X | Motion-X| Motion-X/AIST++/FineDance|AIST++/FineDance | AIST++/FineDance|AIST++/FineDance | AIST++/FineDance|
|Training samples number | 64867 | 64867 | 64867/952/177 |952/177 | 952/177|952/177|952/177|


### **Q3: The results of instruction-tuned only on AIST++ or FineDance (Reviewer #Wkeh)**

 As shown in the tables below, we add the results of $M^3$GPT instruction-tuned only on AIST++ and FineDance. As shown, instruction-tuned on a single task further improves performance. We will add these results in final version.

|Methods on AIST++| Music-to-Dance (FID$_k$$\downarrow$   \|Div$_k$$\uparrow$ \| BAS$\uparrow$)| Dance-to-Music (BCS$\uparrow$ \| BHS$\uparrow$) |
| ---- | ----- | ----- |
$M^3$GPT (Pre-trained) | 27.65 \| 7.52 \| 0.2105 | 93.4 \| 93.8
$M^3$GPT (Instruction-tuned)| 24.34 \| 7.50 \| 0.2056 |  93.6 \| **94.0**|
$M^3$GPT (Instruction-tuned for single task) | **23.01** \| **7.85** \| **0.2261**| **94.3** \| **94.0**

|Methods on FineDance| Music-to-Dance (FID$_k$$\downarrow$   \| Div$_k$$\uparrow$ \| BAS$\uparrow$)| Dance-to-Music (BCS$\uparrow$ \| BHS$\uparrow$) |
| ---- | ----- | ----- |
$M^3$GPT (Pre-trained) | 92.35 \| 7.67 \| 0.2134 | 83.16 \| 73.65|
$M^3$GPT (Instruction-tuned)| 86.47 \| 7.75 \| 0.2158 | 84.10 \| 74.66|
$M^3$GPT (Instruction-tuned only A2D) | 65.27 \| 7.83 \| 0.2195 | 84.79 \| 75.20 |
$M^3$GPT (Instruction-tuned only A2D with 30-second clips) | **42.66** \| **8.24** \| **0.2231** | **86.72** \| **79.64**

### **Q4: The statistics of music2text (A2T) and text2dance (T2D) paired dataset (Reviewer #Wkeh)**

A2T and T2D datasets are constructed from existing dance dataset (AIST++ and FineDance). Given a music-dance pair, we construct paired textural descriptions for the music/dance data.  Specifically, we use the style annotations of music to create paired texts, such as *a person is dancing Jazz* for *Jazz* style. The table below shows the statistics of A2T and T2D datasets. We will added this statistics in final version.

| Task |  Training sample number (AIST++, FineDance) | Testing sample number | Average sequence duration
|----|----|----|----|
| A2T| 1129 (952, 177) | 60 (40, 20) | (13.31s,  135.8s)  |
| T2D| 1129 (952, 177) | 60 (40, 20) | (13.31s,  135.8s) |

---

### Decision · Program_Chairs · 2024-09-25

**Decision:**

Accept (poster)

**Comment:**

This paper introduces $M^3$GPT, a multimodal, multitask framework for motion comprehension and generation, integrating text, music, and motion into a unified representation. The model demonstrates strong performance across tasks and impressive zero-shot generalization capabilities.

Reviewers praised the model’s novelty and its ability to handle multiple input modalities seamlessly within a large language model. However, concerns were raised about the complexity and whether the model’s benefits would generalize to broader tasks. Some reviewers also noted a lack of comparisons with other state-of-the-art frameworks.
The rebuttal provided some clarifications and comparisons, though not all concerns were fully addressed. Despite this, the strong results and novel approach make it a valuable contribution.

Overall, the AC leans towards acceptance, given the paper’s solid contributions to multimodal motion comprehension and generation, while recommending that the authors address the concerns about generalization and further comparisons in future revisions.